# The CHD family chromatin remodeling enzyme, Kismet, promotes both clathrin-mediated and activity-dependent bulk endocytosis

Emily L. Hendricks 📷, Faith L. W. Liebl 📷*

Department of Biological Sciences, Southern Illinois University Edwardsville, Edwardsville, Illinois, United States of America

* fliebl@siue.edu

**Data Availability Statement:** All relevant data are within the manuscript and its Supporting Information files.

## Abstract

Chromodomain helicase DNA binding domain (CHD) proteins, including CHD7 and CHD8, remodel chromatin to enable transcriptional programs. Both proteins are important for proper neural development as heterozygous mutations in *Chd7* and *Chd8* are causative for CHARGE syndrome and correlated with autism spectrum disorders, respectively. Their roles in mature neurons are poorly understood despite influencing the expression of genes required for cell adhesion, neurotransmission, and synaptic plasticity. The *Drosophila* homolog of CHD7 and CHD8, Kismet (Kis), promotes neurotransmission, endocytosis, and larval locomotion. Endocytosis is essential in neurons for replenishing synaptic vesicles, maintaining protein localization, and preserving the size and composition of the presynaptic membrane. Several forms of endocytosis have been identified including clathrin-mediated endocytosis, which is coupled with neural activity and is the most prevalent form of synaptic endocytosis, and activity-dependent bulk endocytosis, which occurs during periods of intense stimulation. Kis modulates the expression of gene products involved in endocytosis including promoting *shaggy/GSK3β* expression while restricting *PI3K92E*. *kis* mutants electrophysiologically phenocopy a *liquid facets* mutant in response to paradigms that induce clathrin-mediated endocytosis and activity-dependent bulk endocytosis. Further, *kis* mutants do not show further reductions in endocytosis when activity-dependent bulk endocytosis or clathrin-mediated endocytosis are pharmacologically inhibited. We find that Kis is important in postsynaptic muscle for proper endocytosis but the ATPase domain of Kis is dispensable for endocytosis. Collectively, our data indicate that Kis promotes both clathrin-mediated endocytosis and activity-dependent bulk endocytosis possibly by promoting transcription of several endocytic genes and maintaining the size of the synaptic vesicle pool.

**Funding:** This work was funded by the National Institute of Neurological Disorders and Stroke of the NIH under award numbers 1R15NS101608-01A1 and 2R15NS101608-02A1 (to FL) and by Southern Illinois University Edwardsville's Competitive Graduate Award (to EH). The funders had no role in study design, data collection and analysis, decision to publish, or preparation of the manuscript.

**Competing interests:** The authors have declared that no competing interests exist.

## Introduction

Neural communication relies on the coordinated release of neurotransmitter, which diffuses across the synaptic cleft and binds to postsynaptic receptors [1, 2]. Synaptic proteins facilitate processes required for proper neurotransmission and synaptic plasticity by inducing cellular changes in response to an action potential [3]. The mobilization and trafficking of synaptic vesicles constitute the presynaptic response to action potential-induced $Ca^{2+}$ influx [4]. Thus, $Ca^{2+}$-dependent release of neurotransmitter-containing vesicles, and the proteins that comprise vesicle release machinery and postsynaptic receptors, are important for mediating neural communication. Presynaptic neurons maintain distinct pools of neurotransmitter-filled vesicles that can be quickly mobilized upon stimulation. The localization of vesicles into these pools influences their temporal availability for release [5, 6]. Vesicles in the readily releasable pool are docked at presynaptic release sites and are the first to be released upon stimulation [7]. The reserve pool, however, may be more distal to sites of release and is mobilized upon high frequency stimulation to replenish the readily releasable pool [8, 9]. Both the readily releasable and reserve pools are replenished by recycling vesicles through endocytosis [10] and disruptions in these endocytic pathways compromise neurotransmitter release.

Endocytosis is a process that involves the internalization and scission of the plasma membrane and functions to recycle membrane-associated proteins and phospholipids. Several mechanisms of endocytosis have been characterized including clathrin-mediated endocytosis (CME), activity-dependent bulk endocytosis (ADBE), kiss-and-run, fast compensatory endocytosis, and ultrafast endocytosis [11, 12]. CME involves the recruitment of clathrin scaffold proteins that surround vesicles upon endocytic uptake and is the primary mode of synaptic vesicle recycling during basal neurotransmission [12, 13]. Conversely, ADBE initially occurs independently of clathrin and is used during high frequency neuronal stimulation to replenish reserve pool vesicles [12, 14]. Both processes facilitate synaptic vesicle recycling and are important for overall neuronal function. Mutations in endocytic machinery-encoding genes are associated with neurodevelopmental disorders [15] and neurodegenerative diseases [16].

The expression of synaptic gene products, including those required for endocytosis, is dynamically regulated by chromatin states. Chromatin regulatory proteins establish and maintain the epigenome, which consists of the cellular chemical modifications of DNA and histones [17]. Epigenetic changes in gene expression are required for synaptic remodeling during learning [18] and occur in both neurodevelopmental disorders [19] and neurodegenerative diseases [20]. The ATP-dependent chromodomain helicase DNA-binding (CHD) family proteins regulate neurodevelopmental processes [21] but are also expressed in mature neurons [22]. Given the partial overlap of misregulated genes shared between autism spectrum disorder and Alzheimer's disease [23], a better understanding of how chromatin remodeling influences synaptic processes during both neurodevelopment and neurodegeneration is warranted. The link between neurodevelopmental and neurodegenerative diseases and the molecular pathways underlying their shared pathology are becoming increasingly clear. For example, mutations in the catalytic γ-secretase subunit, *presenilin*, impair postsynaptic development [24] and are also linked to aberrant proteolytic processing in familial Alzheimer's disease [25]. How dysfunctional CHD proteins specifically impact synaptic function throughout neurodevelopmental and neurodegenerative disease pathways, however, is still poorly characterized.

Kismet (Kis) is the *Drosophila* homolog of group III CHD proteins, including CHD7 and CHD8 [21, 26]. The full length Kis isoform possesses two chromodomains that bind methylated histone residues and an ATPase domain that catalyzes nucleosome remodeling [26, 27]. These nucleosome modifications may activate or repress transcription depending on the binding sites exposed [28]. At the *Drosophila* neuromuscular junction (NMJ), Kis restricts axonal

branching and bouton formation [29] and promotes glutamate receptor localization [30] and synaptic vesicle endocytosis [31]. Here, we find that Kis regulates ADBE and CME at the *Drosophila* NMJ possibly by regulating the expression of endocytosis-related gene products and maintaining vesicle pool sizes. We also show that Kis is important in postsynaptic cells for pre-synaptic vesicle endocytosis and Kis's modulation of endocytosis is independent of ATPase domain activity.

## Materials and methods

### *Drosophila* stocks and husbandry

Fly stocks were maintained at 25˚C with a 12 h light:dark cycle and fed Jazz Mix food (Fisher Scientific AS153). Nutri-Fly Instant Food (Genesee Scientific 66–118) was used for experiments where flies were raised on food containing compounds. Male and female larvae and adults were used for all experiments except for $sgg^{EP1576}$. The $sgg^{EP1576}$ mutation is on the first chromosome necessitating the use of only females of this and the control genotypes. Most fly stocks were obtained from the Bloomington Drosophila stock center including $w^{1118}$ (RRID: BDSC_3605), $dap160^{EP2543}$ (RRID:BDSC_3605), $kis^{k13416}$ (RRID:BDSC_10442), $lqf^{KG03016}$ (RRID:BDSC_13766), $sgg^{EP1576}$ (RRID:BDSC_11008), *Actin5c-Gal4* (RRID:BDSC_30558), $elav^{C155}$-*Gal4* (RRID:BDSC_458), *elav-Gal4* (RRID:BDSC_8760), *D42-Gal4* (RRID: BDSC_8816), *24B-Gal4* (RRID:BDSC_1767), and *repo-Gal4* (RRID:BDSC_7415). $kis^{LM27}$ and *UAS-kis-L* were gifts from Dan Marenda [32]. *UAS-CHD7* and *UAS-kis^{K2060R}* were generated for this work.

### Generation and expression of *UAS-CHD7* and *UAS-kis^{K2060R}*

Human CHD7 (NM_017780) was used to determine the *Drosophila* optimized *CHD7* cDNA sequence. The *kis-L* (NM_001258889) cDNA sequence was used for *kis^{K2060R}* but was modified by substituting CGA (encoding R) for AAA (encoding K) at codon 2059. cDNAs were synthesized by Thermo Fisher and included a *NotI* restriction site on the 5' end and a *XbaI* site on the 3' end. A *Drosophila* optimized V5 cDNA sequence (`GGT AAG CCC ATC CCG AAC CCC CTG CTG GGT TTG GAC TCC ACT`) was included upstream of the stop codon and *XbaI* restriction site. cDNAs were inserted into *pUAST* using the *NotI* and *XbaI* restriction sites and insertion was confirmed by DNA sequencing. Both constructs were injected into $w^{1118}$ embryos using standard germline transformation methods (BestGene, Inc., Chino Hills, CA).

### Reverse transcription PCR and qPCR

Central nervous systems (CNSs) were dissected from male and female third instar larvae in Roger's Ringer solution (135 mM NaCl, 5 mM KCl, 4 mM MgCl$_2$*6H$_2$O, 1.8 mM CaCl$_2$*2H$_2$O, 5 mM TES, 72 mM Sucrose, 2 mM glutamate, pH 7.15), placed in Invitrogen RNAlater Stabilization Solution (Fisher Scientific AM7020), and stored at -20˚C. 30 CNSs or 8–12 larvae per genotype were used for each technical replicate. RNA was isolated using the Invitrogen Purelink RNA Mini Kit (Fisher Scientific 12-183-025). RNA concentrations were obtained using an Implen Nanophotometer N50. 100 ng of RNA was used for each reaction. Primers were designed using PerlPrimer (v. 1.1.21). qRT-PCR was performed using the iTaq Universal SYBR Green One Step Kit (Bio-Rad 1725151) and a CFX Connect Real-Time PCR Detection System (Bio-Rad). At least four biological replicates each including three technical replicates were used for data analysis. $2^{-\Delta\Delta C(t)}$ values [33] were calculated by first subtracting the C(t) value of the target transcript reaction from the C(t) value for *GAPDH* to obtain $\Delta C(t)$ for each transcript. Next, the difference between control and *kis* mutant $\Delta C(t)$s was calculated

to obtain the $2^{-\Delta\Delta C(t)}$. $2^{-\Delta\Delta C(t)}$ were calculated using RNA samples isolated the same day with RT-qPCR reactions executed simultaneously. Student's t-tests were used to determine if there was a statistical difference between control and *kis* mutant $\Delta C(t)$s.

## Immunocytochemistry and FM labeling

Third instar larvae were fillet dissected in Roger's Ringer solution (135 mM NaCl, 5 mM KCl, 4 mM MgCl$_2$*6H$_2$O, 1.8 mM CaCl$_2$*2H$_2$O, 5 mM TES, 72 mM Sucrose, 2 mM glutamate, pH 7.15) on Sylgard (World Precision Instruments)-coated 60 mm dishes. Larvae were fixed for 30 min with 4% paraformaldehyde (Fisher Scientific F79500) in 1 x phosphate buffered saline (PBS, Midwest Scientific QS1200). Fixed larvae were placed in 1.5 ml centrifuge tubes containing PTX (1 x PBS + 0.1% Triton X-100, Fisher Scientific AAA16046AP) and washed three times for 10 min in PTX. After two 30 min washes in PBTX (1 x PBS + 0.1% Triton X-100 + 1% Bovine Serum Albumin, Fisher Scientific BP1600-100), rabbit α-V5 (1:1000, Sigma AB3792, RRID: AB_91591) was diluted in PBTX and applied overnight at 4˚C. After primary antibodies were removed, larvae were washed three times for 10 min and two times for 30 min in PBTX. α-rabbit FITC (Jackson ImmunoResearch; RRID: AB_2337972) was diluted 1:400 in PBTX and applied for 2 h at room temperature with Cy3 HRP (1:125, Jackson ImmunoResearch, RRID: AB_2338959) and DAPI (1:500, ThermoFisher D1306, RRID: AB_2629482).

Endocytosis was measured as described by Verstreken et al. [34]. Briefly, third instar larvae were fillet dissected in Ca$^{2+}$-free HL-3 (100 mM NaCl, 5 mM KCl, 10 mM NaHCO$_3$, 5 mM HEPES, 30 mM sucrose, 5 mM trehelose, 10 mM MgCl$_2$, pH 7.2). After one wash with Ca$^{2+}$-free HL-3 and cutting the motor neurons, 4 μM FM 1-43FX (Fisher Scientific F35355) in HL-3 containing 90 mM KCl and 1.0 mM CaCl$_2$ was applied for one minute. For endocytosis inhibition experiments, larvae were exposed to DMSO, 100 μM BAPTA-AM for 10 min, 25 μM EGTA-AM for 10 min, 200 μM Chlorpromazine for 30 min, 100 μM Dynasore for 10 min, or 200 μM Roscovitine for 30 min prior to one min stimulation with 90 mM KCl. Concentrations and treatment times were determined using previously published protocols (see results). Next, larvae were washed five times over 5–10 min in Ca$^{2+}$-free HL-3. Larvae were fixed in 3.7% paraformaldehyde (Fisher Scientific) in Ca$^{2+}$-free HL-3 for 5 min and then placed in 1.5 ml centrifuge tubes. Larvae were washed with Ca$^{2+}$-free HL-3 containing 2.5% goat serum several times and with Ca$^{2+}$-free HL-3 10 times over 10–15 min. A647 HRP (1:125, Jackson ImmunoResearch) was applied for 30 min in Ca$^{2+}$-free HL-3 containing 5% goat serum. Next, larvae were washed with Ca$^{2+}$-free HL-3 10 times over 10–15 min and mounted on slides with Vectashield (Vector Laboratories H1000).

An Olympus FV1000 confocal microscope was used to acquire images of 6/7 NMJs within segments 3 or 4 and ventral nerve cords. The former were acquired using a 60x oil immersion objective while the latter were acquired using a 40x oil immersion objective. Genotypes were labeled using the same reagents for each experimental replicate. The mean of all control acquisition settings was used for each experimental animal and approximately equal numbers of controls and experimental animals were imaged each day. Image z-stacks were constructed using Fiji [35]. Measurement of FM 1-43FX signal intensities was accomplished by calculating the relative fluorescence intensity, which was the difference between the synaptic fluorescence and background fluorescence, from each max projected z-stack slice. The mean relative fluorescence value of all slices was used as the NMJ fluorescence intensity for each NMJ and this is represented as a point on bar graphs.

## Electrophysiology

Third instar larvae were fillet dissected in ice cold HL-3 containing 0.25 mM $Ca^{2+}$ and glued to Sylgard (World Precision Instruments)-coated round 18 mm coverslips. After the nervous system was removed, larvae were rinsed once with room temperature HL-3 containing 1.0 mM $Ca^{2+}$, which was used for recordings. Muscle 6 of body wall segments 3 or 4 was used for two electrode voltage clamp recordings. Electrodes were filled with 3 M KCl and used for recordings provided they demonstrated resistances of 10–30 MΩ. Recordings were acquired in pClamp (Molecular Devices, v. 11.1) and obtained from muscles clamped at -60 mV using an Axoclamp 900A amplifier (Molecular Devices) with input resistances <5 MΩ. Segmental nerves were stimulated with suction electrodes filled with HL-3 containing 1.0 mM $Ca^{2+}$. Suprathreshold stimuli were administered from a Grass S88 stimulator with a SIU5 isolation unit (Grass Technologies). High frequency stimulation included stimuli administered at 0.2 Hz for 50 s, 20 Hz for 60 s, and 0.2 Hz for 50 s. Low frequency stimulation included 5 Hz stimulation for five minutes. Stock concentrations of each compound were diluted to 100 μM BAP-TA-AM, 25 μM EGTA-AM, and 100 μM Dynasore in HL-3 containing 1.0 mM $Ca^{2+}$ the day of recording and kept on ice. Working concentrations were allowed to warm to room temperature immediately before use. Vesicle pool sizes were assessed by incubating dissected larvae at room temperature for 20 min in freshly prepared 2 μM Bafilomycin in HL-3 containing 1 mM $Ca^{2+}$. Concentrations and treatment times for all compounds were determined using previously published protocols (see results). mEJCs were recorded for 3 min followed by stimulation of the segmental nerve at 3 Hz for 10 min or at 10 Hz for 5 min. Recordings were digitized with a Digidata 1443 digitizer (Molecular Devices).

Recordings from approximately the same number of controls and experimental animals were obtained each day. All experiments included at least three biological replicates with sample sizes of 9–11 larvae as indicated in figure legends. Data were analyzed in Clampfit (v 11.1, Molecular Devices) and GraphPad Prism (v. 9.3.0). When trains of stimulation were delivered, eEJC amplitudes were normalized to the first eEJC. Multiple unpaired t-tests were used to determine if eEJC amplitudes differed over time.

## Behavior

Third instar larvae were obtained from vials with Jazz Mix food and briefly wiped with a damp paintbrush to remove any debris. Larvae were then transferred to a 1.6% agar plate and allowed to freely wander for one minute to acclimate to crawling surface. After the acclimation period, larvae were moved to the behavior arena (a backlit 1.6% agar slab with overhead camera mount) and video recorded for 30 seconds with a Cannon EOS M50 camera at 29.97 frames per second. Recording started after larvae engaged in forward-directed motion. 899 frames were analyzed in Fiji (NIH ImageJ) with the wrMTrck plugin by Jesper S. Pedersen [36] to obtain values for average distances travelled, average and maximum velocities, and body lengths per second. XY coordinates of larval crawling paths were extracted from wrMTrck and graphed in Excel (Microsoft) to generate representative path tracings.

## Experimental design and statistical analyses

All experiments included at least two biological replicates with approximately equal numbers of controls and experimental animals. Data analyses were performed with GraphPad Prism (v. 9.3.0 and 10.0.0). When data sets included one control group, for example comparison of mutant genotypes to $w^{1118}$ controls, means were compared using unpaired t-tests. When data sets included two control groups, a one-way ANOVA was used followed by Tukey's post hoc tests. Bartlett's Test for homogeneity of variance was used to assess the variances between data

sets. For experiments that included more than one outcrossed *Gal4* control groups, the control groups were compared to determine if they statistically differed. If the *Gal4* control groups did not statistically differ, they were combined and treated as a single control group. Bar graphs in figures represent the means and show sample sizes as individual points. Sample sizes indicate individual larvae except for the RT-qPCR experiments where the points represent one technical replicate. Statistical significance is represented on bar graphs as follows: * = <0.05, ** = <0.01, *** = <0.001 with error bars representing standard error of the mean (SEM). S1 Table shows the statistically significant p-values corresponding to each figure.

## Results

### Kismet influences transcription of gene products encoding CME- and ADBE-associated proteins

Presynaptic endocytosis replenishes synaptic vesicles [37], influences protein localization, and preserves the size and composition of the presynaptic membrane [38, 39]. We previously showed that that Kis promotes endocytosis by regulating transcription of endocytic genes and synaptic Dynamin (Dyn) localization. Specifically, mutations in *kis* increased *AP2α* but decreased *dap160* and *endophilin B* (*endoB*) transcripts [31]. Each of these gene products is required for CME. Dap160/Intersectin is a scaffolding protein that recruits the clathrin adapter AP2 to the membrane [40] while endophilins recruit the GTPase, Dyn [41], and help induce curvature of the plasma membrane [42]. In this study, we sought to better understand the role of Kis in the distinct regulation of CME, ADBE, or both using the *Drosophila* NMJ. The *Drosophila* NMJ provides a well-established system to model mammalian neurotransmission as these synapses share common structural features and mechanisms of neurotransmitter release, vesicle endocytosis, and postsynaptic signaling [43].

To further investigate Kis' regulation of endocytosis, we extended our analysis to gene products involved in CME, ADBE, or both. We used two *kis* mutant alleles for our analysis including the hypomorphic allele, *kis^{k13416}*, and *kis^{LM27}*, a null allele [29]. Because the latter is embryonic lethal, we used animals heterozygous for *kis^{k13416}* and *kis^{LM27}* to examine endocytic transcript levels via RT-qPCR. Of the 10 transcripts examined, *PI3K92E*, which encodes a catalytic subunit of the phosphatidylinositol 3-kinase (PI3K) enzyme, was increased in *kis^{LM27}*/*kis^{k13416}* but not *kis^{k13416}* mutants compared with controls. In contrast, *shaggy (sgg)*, which encodes Glycogen Synthase Kinase 3 (GSK3) was decreased in both *kis* mutants relative to *w^{1118}* controls (Fig 1). PI3K indirectly inhibits GSK3 activity thereby activating ADBE [44]. These data, taken together with our previous observations [31], suggest that Kis may regulate the expression of several CME and ADBE transcripts.

### Kismet promotes CME and ADBE

To determine whether Kis is functionally involved in CME and ADBE, we assessed endocytosis via electrophysiology and uptake of the lipophilic dye FM 1-43FX. We first used a 20 Hz high frequency stimulation (HFS) protocol in 1.0 mM $Ca^{2+}$ for one minute to promote ADBE [45, 46]. This was followed by 0.2 Hz stimulation to assess recovery, which requires the readily releasable pool of vesicles to be replenished by mobilization of the reserve pool [47]. We also assessed endocytosis in *liquid facets* (*lqf*) mutants because of its roles in CME and ADBE. Lqf, which is the *Drosophila* homolog of Epsin 1 [48], remodels the membrane and recycles synaptic vesicles during CME [37] and is dephosphorylated along with several other proteins to trigger ADBE [49]. Both *kis* and *lqf* mutants showed decreased evoked endplate junctional current (eEJC) amplitudes during HFS compared to controls (Fig 2A and 2B). Recovery from HFS was

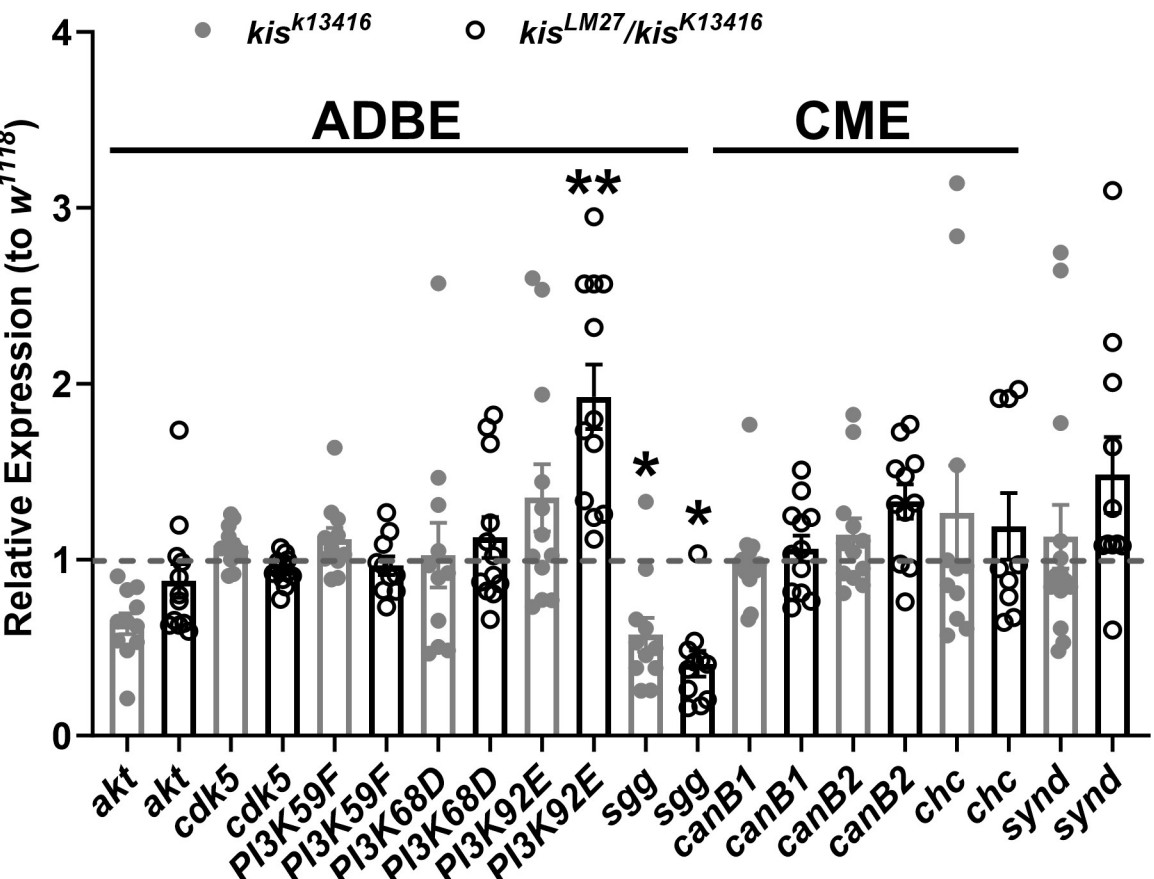

**Fig 1. Kismet affects the expression of gene products involved in both CME and ADBE.** *PI3K92E* and *shaggy/GSK3β* (*sgg*) transcripts are differentially expressed in *kis* mutants. Relative expression of CNS transcripts was assessed via RT-qPCR. $2^{-\Delta\Delta C(t)}$ values are indicated. Data includes at least four biological replicates each including three technical replicates. Technical replicates are represented by circles for the representative genotypes. Bars indicate means with the error bars showing the SEM.

impaired in *kis* mutants from 40–50 seconds after HFS and at 40 seconds after HFS in *lqf* mutants (100–110 seconds in Fig 2A). These data indicate that *kis* and *lqf* mutants show impaired ADBE and recycling of vesicles after HFS.

Next, we used a low frequency stimulation (LFS) protocol consisting of five minutes of 5 Hz stimulation in 1.0 mM $Ca^{2+}$ to promote CME [50] in *kis* and *lqf* mutants. eEJC amplitudes in controls were approximately 80% of the initial stimulus amplitudes 10–50 seconds after the onset of stimulation. Amplitudes continued to decline until they were approximately 67% of the initial stimulus at the end of stimulation. Although *lqf^{k03016}* mutant eEJC amplitudes were reduced to approximately 76% of the initial stimulus at the beginning and 57% at the end of stimulation, there was only a significant reduction in eEJC amplitudes at 170 seconds after onset of stimulation (Fig 3A and 3B). *kis^{LM27}/kis^{k13416}* mutants exhibited a similar reduction as *lqf^{k03016}* mutants in initial eEJC amplitudes, which were approximately 72% of the first stimulus. There was a significant reduction in eEJC amplitudes in *kis* mutants at several time points from 160–260 seconds after the onset of stimulation when eEJC amplitudes were approximately 60–62% of the initial stimulus (Fig 3A and 3B). Reduced eEJC amplitudes in *kis* mutants when CME was induced suggests that Kis promotes CME.

If Kis regulates CME or ADBE, we would expect to observe no change in eEJC amplitudes during stimulation when CME or ADBE is inhibited. We used several putative inhibitors to

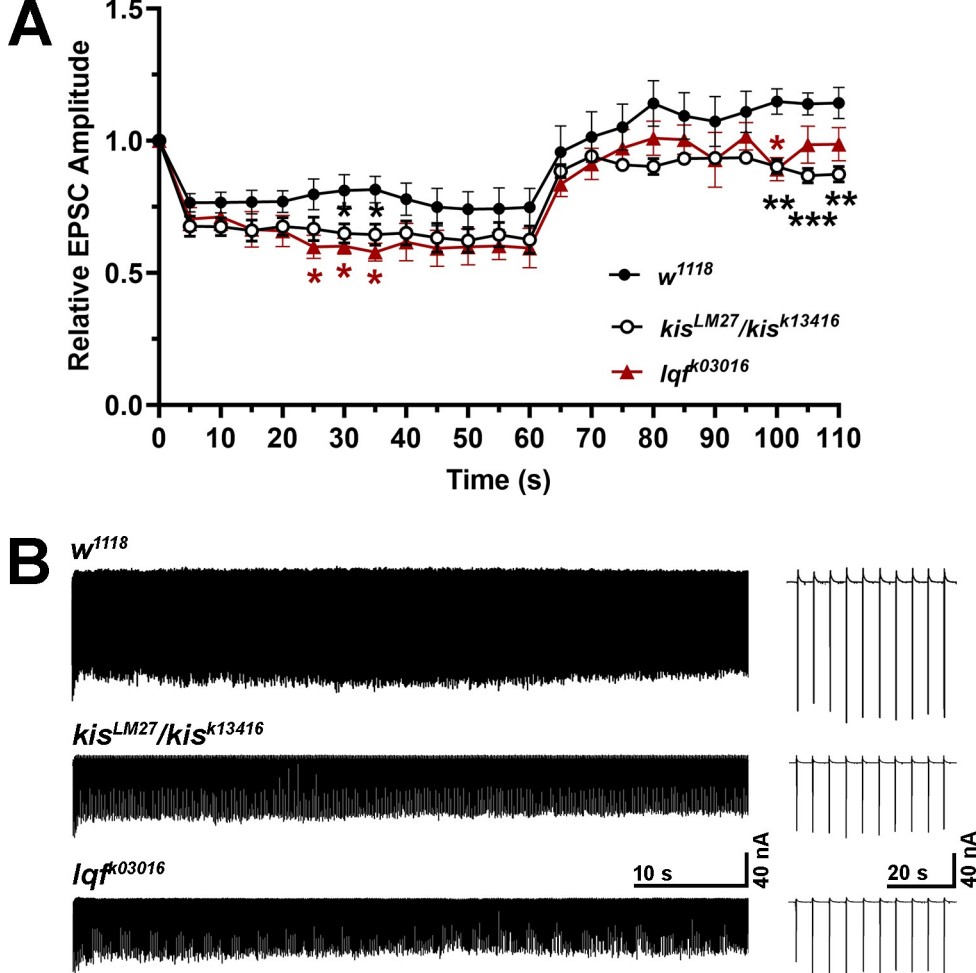

**Fig 2. Kismet and Liquid Facets promote ADBE.** eEJCs were measured during 60 sec of 20 Hz HFS to induce ADBE and during a 50 sec recovery period with 0.2 Hz stimulation. (A) Mean eEJC amplitudes are shown relative to the amplitude measured after the first stimuli for $w^{1118}$ controls (n = 10) and *kis* (n = 9) and *lqf* (n = 12) mutants. Error bars represent the SEM. (B) Representative eEJC recordings from the genotypes listed.

target CME or ADBE in *kis* and *lqf* mutants. CME and ADBE require presynaptic increases in $Ca^{2+}$ [51, 52]. Therefore, we used the $Ca^{2+}$ chelators BAPTA, to inhibit both CME and ADBE, and EGTA, to inhibit ADBE [53]. CME is triggered by local increases in active zone $Ca^{2+}$ initiated by the action potential [54] while ADBE is triggered when $Ca^{2+}$ diffuses away from the active zone to the periactive zone [55]. BAPTA and EGTA are $Ca^{2+}$ chelators that inhibit increases in $Ca^{2+}$ at the periactive zone. Only BAPTA, however, inhibits the increases in $Ca^{2+}$ at the active zone [56] because it has a faster on binding rate [57]. We also used Dynasore, an inhibitor of Dyn GTPase activity [58], to block CME. Dynasore does not affect Dyn's affinity for GTP or its capacity to assemble on membranes [59]. After a 10 minute incubation, 100 μM Dynasore inhibited CME induced by LFS (S1 Fig) but did not affect ADBE (S2 Fig) in $w^{1118}$ controls. The most pronounced effect on CME (S1 Fig) was observed from 160–250 sec. During this time, control eEJCs were 75.6–77.7% of the first response. In contrast, Dynasore treatment produced eEJCs that were 57.8–64.1% of the first response. Neither 100 μM BAPTA-AM nor 25 μM EGTA-AM affected CME or ADBE (S1 and S2 Figs) in controls after a 10 minute incubation. Therefore, we used Dynasore to determine if Kis functionally influences CME.

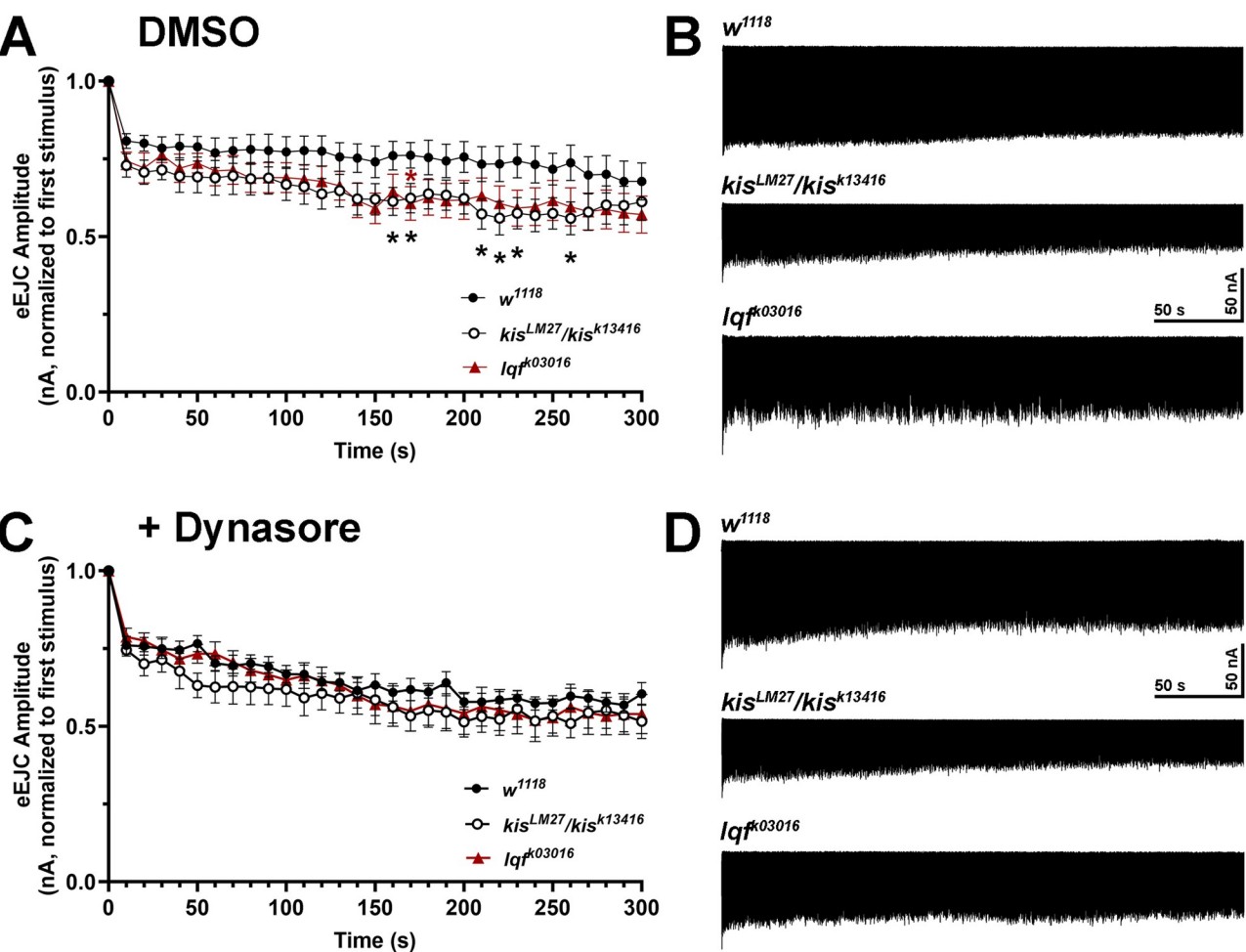

**Fig 3. Inhibition of CME using Dynasore does not produce a change in evoked currents in *kis* or *lqf* mutants.** eEJCs were measured during five min of 10 Hz stimulation to induce CME. (A) Mean eEJC amplitudes are shown relative to the amplitude measured after the first stimuli for *w^1118^* controls (n = 9) and *kis* (n = 10) and *lqf* (n = 10) mutants. Error bars represent the SEM. (B) Representative eEJC recordings from the genotypes listed. (C) eEJC amplitudes were obtained after animals were pretreated with 100 μM Dynasore. Mean eEJC amplitudes are shown relative to the amplitude measured after the first stimuli for *w^1118^* controls (n = 9) and *kis* (n = 10) and *lqf* (n = 10) mutants. Error bars represent the SEM. (D) Representative eEJC recordings from the genotypes listed after pretreatment with 100 μM Dynasore.

Both *kis* and *lqf* mutants exhibited reduced eEJC amplitudes during LFS similar to that of controls after a 10 minute incubation with 100 μM Dynasore (Fig 3C and 3D). The similar reduction in LFS-induced eEJCs when CME was inhibited with Dynasore suggests that Kis and Lqf contribute to CME.

We extended our analyses of CME and ADBE to include additional inhibitors and endocytic mutants as negative controls. In addition to the *lqf^KG03016^* mutant, which exhibits alterations in both CME and ADBE [37, 60], we used *dap160^EP2543^* and *sgg^EP1576^* mutants, which have impaired CME [40] and ADBE [44], respectively. Endocytosis was examined using the lipophilic dye, FM 1–43FX, to label newly endocytosed synaptic vesicles [34] after one min stimulation with 90 mM KCl in 1.0 mM Ca²⁺. Stimulation using high K⁺ induces both CME and ADBE [61]. This stimulation paradigm resulted in decreased endocytosis, as indicated by a significant reduction in FM 1-43FX internalization, in all four mutants (Fig 4B, left). Next, larvae were pretreated with either 100 μM BAPTA-AM, 25 μM EGTA-AM, or 200 μM Roscovitine before one min stimulation with 90 mM KCl in 1.0 mM Ca²⁺. Each compound pretreatment

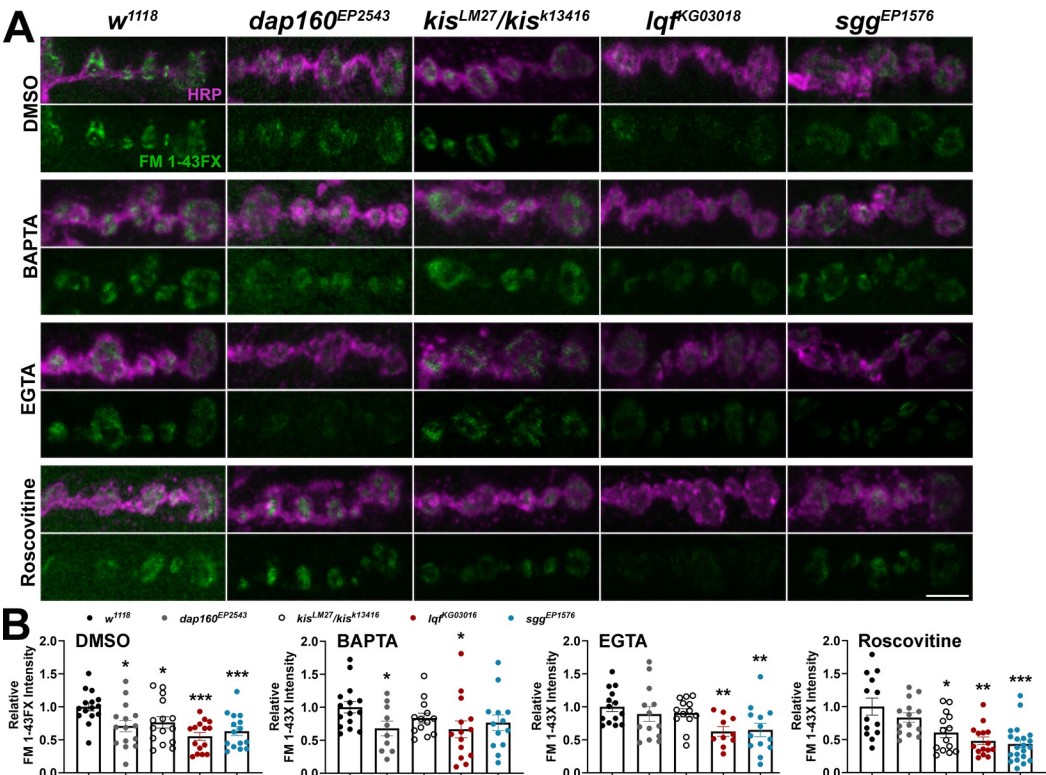

**Fig 4.** *Kismet* **mutants exhibit minimal changes in endocytosis when ADBE is inhibited.** Endocytosis was assessed by measuring internalization of the lipophilic dye FM 1-43FX after one min stimulation with 90 mM KCl. Animals were pretreated with either 100 μM BAPTA-AM, 25 μM EGTA-AM, or 200 μM Roscovitine to inhibit ADBE. (A) Panels show high resolution confocal micrographs of terminal presynaptic motor neuron boutons (HRP, magenta) after internalization of the lipophilic dye FM 1-43FX (green). (B) Data for each condition were normalized to $w^{1118}$ controls. Bars indicate means, points represent individual larvae, and error bars represent SEM. Endocytic and *kis* mutants show reduced endocytosis. Endocytosis was unchanged in *kis* mutants pretreated with BAPTA or EGTA but reduced when pretreated with Roscovitine. Scale bar = 5 μm.

led to a reduction in endocytosis in $w^{1118}$ controls (S3 Fig). Roscovitine inhibits cyclin-dependent kinase 5, which is required for ADBE [62], by competing with ATP for the kinase domain [63]. Mutations in either *dap160* or *kis* did not affect endocytosis when NMJs were pretreated with BAPTA or EGTA (Fig 4A, middle panels). Similar as $lqf^{KG0301}$ and $sgg^{EP1576}$ mutants, however, $kis^{LM27}/kis^{k13416}$ mutants exhibited reduced endocytosis after pretreatment with the ADBE inhibitor, Roscovitine (Fig 4B). Given that *kis* mutants did not exhibit a further reduction in endocytosis when ADBE was inhibited by BAPTA or EGTA, Kis likely functionally influences ADBE.

We used the same protocol to assess whether Kis also functionally contributes to CME. We again used 100 μM Dynasore to inhibit CME. We also employed Chlorpromazine, which inhibits assembly of clathrin coats [64] possibly by interfering with Dyn-lipid binding [65]. Pretreatment with each compound led to a reduction in endocytosis $w^{1118}$ controls (S3 Fig). None of the mutants examined exhibited changes in endocytosis after pretreatment with 200 μM Chlorpromazine while only $sgg^{EP1576}$ mutants exhibited a further decrease in endocytosis after pretreatment with 100 μM Dynasore (Fig 5). These data, coupled with our electrophysiological analyses, suggest that Kis functionally contributes to both CME and ADBE.

Reduced eEJC amplitudes during LFS and HFS and endocytosis could be a consequence of *kis* mutant synapses containing fewer synaptic vesicles. To test this possibility, we used the

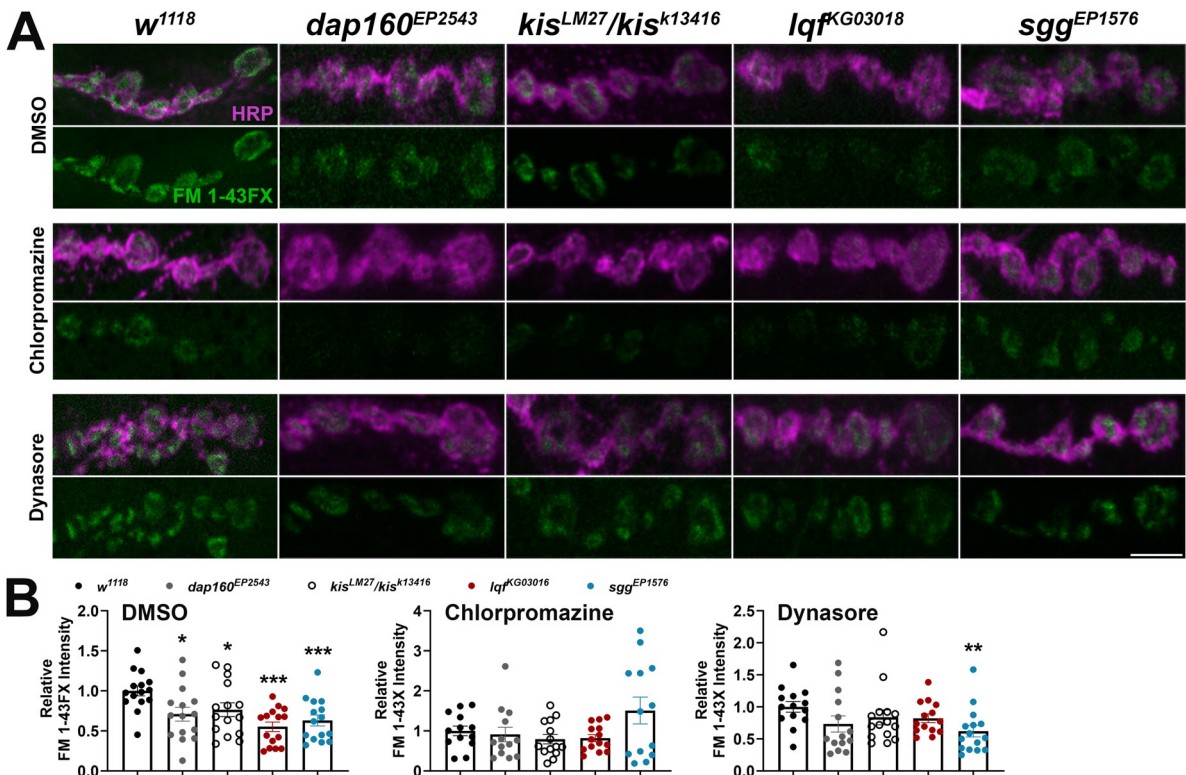

**Fig 5.** *Kismet* **mutants do not show changes in endocytosis when CME is inhibited.** Endocytosis was assessed by measuring internalization of the lipophilic dye FM 1-43FX after one min stimulation with 90 mM KCl. Larvae were pretreated with either 200 μM Chlorpromazine or 100 μM Dynasore to inhibit CME. (A) Panels show high resolution confocal micrographs of terminal presynaptic motor neuron boutons (HRP, magenta) after internalization of the lipophilic dye FM 1-43FX (green). (B) Data for each condition were normalized to $w^{1118}$ controls. Bars indicate means, points represent individual larvae, and error bars represent SEM. Endocytosis was unchanged in *kis* mutants pretreated with Chlorpromazine or Dynasore. Scale bar = 5 μm.

vesicular H$^+$ pump inhibitor, Bafilomycin A1, to block glutamate uptake into vesicles [66]. Because Bafilomycin inhibits the refilling of glutamate into newly endocytosed vesicles, eEJC amplitudes diminish upon successive stimulation as less glutamate is released. eEJC amplitudes were assessed after 20 min incubation with 2 μM Bafilomycin in 1.0 mM Ca$^{2+}$ during 3 or 10 Hz stimulation. 3 Hz stimulation promotes vesicle release from the readily releasable and recycling vesicle pools while 10 Hz stimulation mobilizes the reserve pool of vesicles [67]. *kis* mutants exhibited a more pronounced reduction in eEJC amplitudes relative to the first stimuli during both 3 and 10 Hz stimulation compared with $w^{1118}$ controls (Fig 6). These data indicate *kis* mutant synapses possess fewer vesicles.

## Kismet is important in postsynaptic muscles for endocytosis

We performed rescue experiments to determine the tissue-specific requirements of Kis for endocytosis and locomotor behavior. We restored *kis* expression in $kis^{LM27}/kis^{k13416}$ mutants by expressing *UAS-kis-L* in all tissues using the *Actin5c-Gal4* driver, neurons using the *elav-Gal4* driver, postsynaptic muscle using the *24B-Gal4* driver, or glial cells using the *repo-Gal4* driver. Endocytosis was induced by one min stimulation with 90 mM KCl in HL-3 containing 1.0 mM Ca$^{2+}$ [34]. Expression of *kis-L* in all tissues restored endocytosis (S4A and S4B Fig). There were slight but significant reductions in endocytosis, as indicated by reduced FM 1-43FX internalization, when *kis-L* was expressed in neurons or glia of *kis* mutants compared

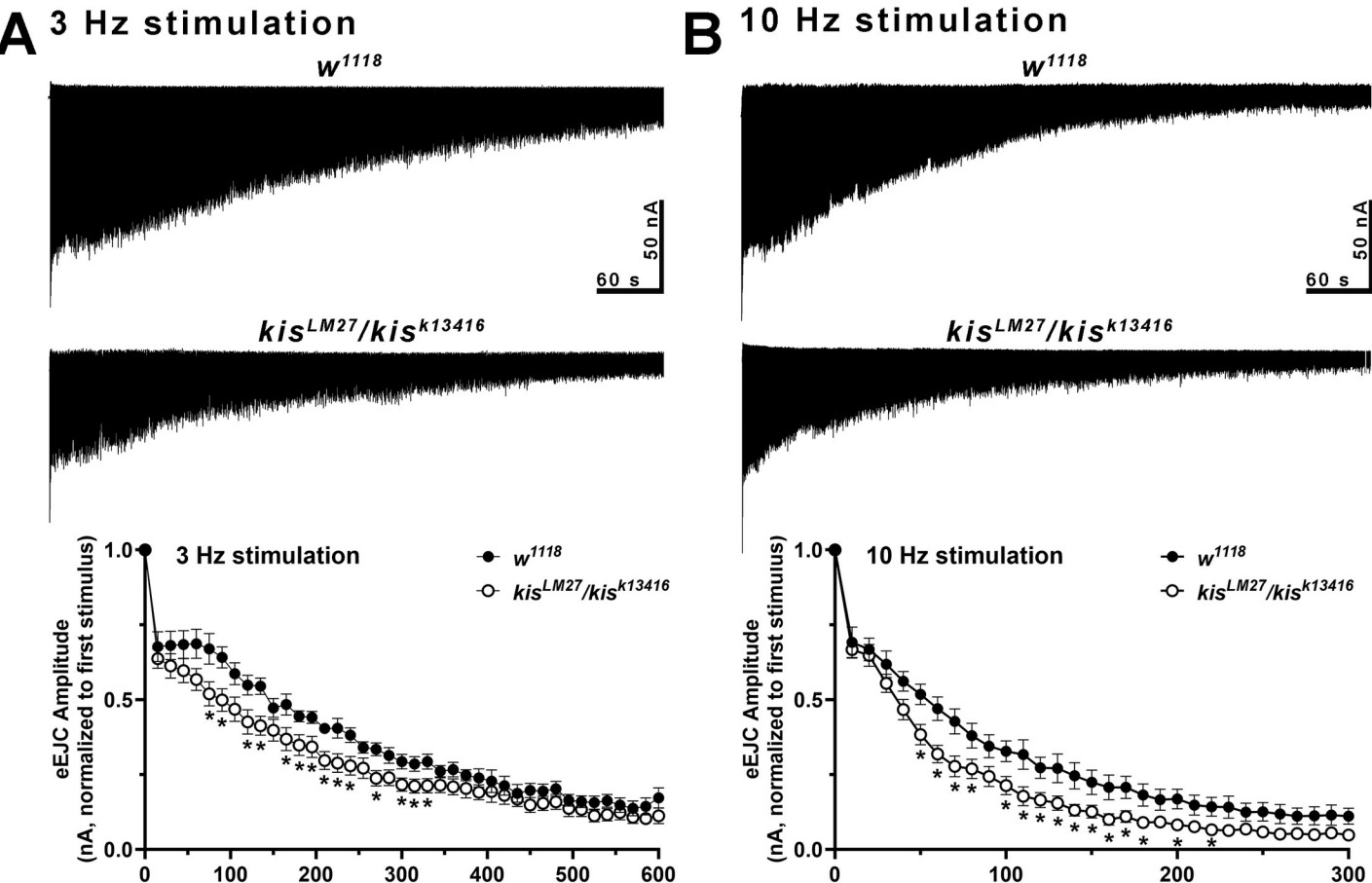

**Fig 6. Kismet helps maintain vesicle pools.** Representative responses (top) in controls (n = 9) and *kis* mutants (n = 10) to 3 (A) or 10 (B) Hz stimulation in the presence of 2 μM Bafilomycin, which inhibits vesicle refilling. Muscles were clamped at -60 mV and recordings were obtained in HL-3 + 1.0 mM $Ca^{2+}$. 3 Hz stimulation depletes the readily releasable and recycling pool of vesicles while 10 Hz stimulation deplete the reserve pool of vesicles. Vesicle pools are smaller in *kis* mutants as indicated by reduced eEJCs induced by stimulation. Bottom graphs show mean eEJC amplitudes normalized to the first stimulus for each condition. Error bars represent the SEM.

with *UAS* but not driver-specific outcrossed controls (Fig 7A and 7B). When *kis-L* was expressed in muscles of *kis* mutants using the *24B-Gal4* driver, however, endocytosis was increased compared with the muscle-specific outcrossed control. There was no difference in endocytosis between *kis* mutants expressing *kis-L* in postsynaptic muscle and *UAS-kis-L/+* outcrossed controls (Fig 7A and 7B). These data indicate that Kis specifically functions postsynaptically to promote endocytosis as postsynaptic *kis-L* expression in *kis^LM27^/kis^k13416^* mutants partially rescues endocytic deficits.

To further probe the tissue-specific contributions of Kis, we knocked down Kis in all tissues, neurons, or postsynaptic muscle to determine if we could mimic the reduction in endocytosis in *kis* mutants. Expression of *UAS-kis^RNAi.b^* was previously shown to reduce Kis levels by 90% in third instar wing discs [29] and reduces *kis* transcripts by 56.2% in all tissues (S4C Fig). When Kis was knocked down in all tissues using the *Actin5c-Gal4* driver, the intensity of the FM signal was decreased to approximately 76.7% of control levels. Similarly, knock down of Kis in postsynaptic muscles using the *24B-Gal4* driver, resulted in a 76.8% decrease in endocytosis (Fig 7C). These values are similar to those of *kis^LM27^/kis^k13416^* mutants, which exhibit a 72.6% decrease in endocytosis. There was no change in endocytosis when Kis was knocked

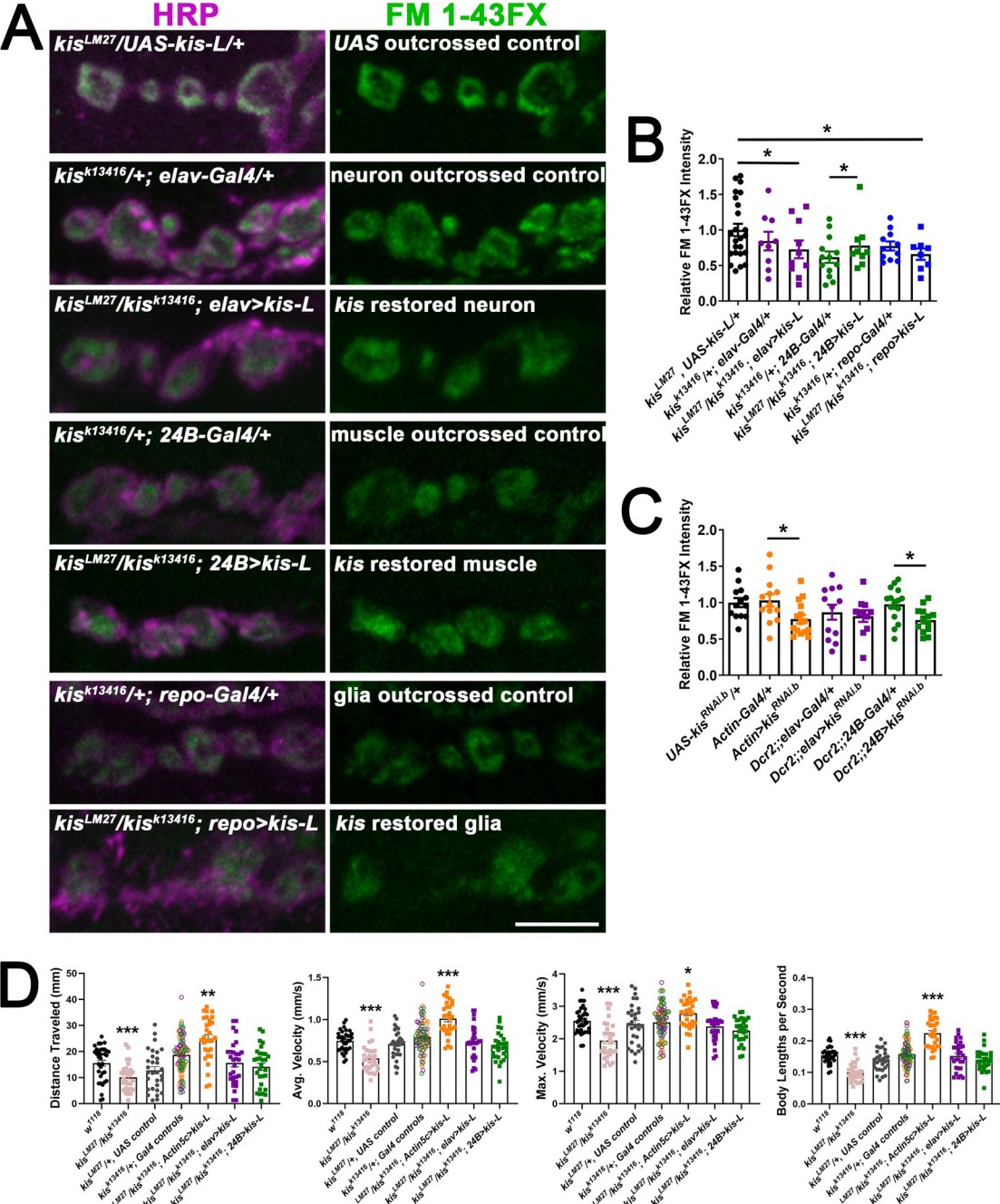

**Fig 7. Kismet is required in postsynaptic muscles for proper endocytosis of presynaptic vesicles.** Tissue-specific drivers were used to restore *kis* expression in specific tissues of *kis* mutants including presynaptic motor neurons (using the *elav-Gal4* driver), postsynaptic muscle (using the *24B-Gal4* driver), or glial cells (using the *repo-Gal4* driver). A) High resolution confocal micrographs of terminal presynaptic motor neuron boutons (HRP, magenta) after internalization of the lipophilic dye FM 1-43FX (green) after one min stimulation with 90 mM KCl in genotypes as listed. Scale bar = 5 µm. B) Quantification of FM 1-43FX fluorescence intensity indicates that when *kis* is restored in postsynaptic muscles of kis mutants, presynaptic endocytosis returns to control levels. C) Quantification of FM 1-43FX fluorescence intensity after tissue-specific knockdown of Kis. Knockdown of Kis in all tissues using the *Actin5c-Gal4* driver or in postsynaptic muscles using the *24B-Gal4* driver resulted in impaired endocytosis. D) Histograms show quantification as determined by wrMTrck of larval crawling behavior on an agar arena for 30 s. Distance traveled, maximum velocity, average velocity, and crawling velocity normalized to body size (body lengths per second). Restoring *kis* expression in all tissues led to enhanced larval crawling.

down in neurons (Fig 7C). These data support the hypothesis that postsynaptic Kis regulates presynaptic endocytosis.

*Kis* mutants show impaired locomotor behaviors with reductions in the velocity of movement and total distance traveled (Fig 7D). Expression of *UAS-kis-L* in all tissues of $kis^{LM27}$/$kis^{k13416}$ mutants using the *Actin5c-Gal4* driver resulted in increases in total distance traveled and both the average and maximum velocity of larval movement (Fig 7D). In contrast, expression of *UAS-kis-L* in neurons or muscles of $kis^{LM27}$/$kis^{k13416}$ mutants did not affect larval movement compared with controls (Fig 7D). These data indicate that Kis functions in all tissues to promote proper larval locomotion.

## Expression of human CHD7 in *kis* mutants restores locomotion but not endocytosis

Kis is 63% identical to human CHD7 with notable conservation within all major functional domains [26]. CHD7 regulates the transcription of gene products that promote migration of neural crest cells [68] but is also expressed in human adult cortical neurons [22]. Because CHD7 influences the expression of genes required for cell adhesion, neurotransmission, and synaptic plasticity [69], we sought to determine whether expression of human CHD7 in *kis* mutants would restore endocytosis and locomotion. We expressed a *Drosophila* optimized human *CHD7* in all tissues, neurons, or postsynaptic muscles in $kis^{LM27}$/$kis^{k13416}$ mutants after verifying that CHD7-V5 was expressed in the nuclei of these tissues (S5A Fig). Although there was some extranuclear/cytoplasmic V5 signal, the most intense signal was localized to the nucleus. Transgenic expression of CHD7 was accomplished in all tissues using the *Actin5c-Gal4* driver, in neurons using the *elav-Gal4* driver, or in postsynaptic muscle using the *24B-Gal4* driver in $kis^{LM27}$/$kis^{k13416}$ mutants. Expression of CHD7 in all three tissue types restored the maximum velocity of larval movement and distance traveled (Fig 8A and 8B). CHD7 expression in all tissues produced a significant increase in average velocity of larval movement while CHD7 expression in neurons or postsynaptic muscle did not restore this aspect of behavior (Fig 8A and 8B). Conversely, expression of CHD7 failed to restore endocytosis as indicated by internalization of FM 1-43FX (Fig 8C and 8D). Expression of CHD7 in all tissues or postsynaptic muscle instead resulted in reduced endocytosis compared with the *UAS* control but not the *Gal4* control (Fig 8C and 8D). These data indicate that CHD7 restores locomotion, but not endocytosis, in $kis^{LM27}$/$kis^{k13416}$ mutants.

## Expression of an ATPase deficient Kis in *kis* mutants rescues endocytosis and behavior

If the chromatin remodeling activity of Kis is important for endocytosis, we would expect that expression of an ATPase deficient Kis in *kis* mutants would fail to rescue either endocytosis and/or behavior. We mutated the Lys residue within the conserved ATP binding site [70] to an Arg ($Kis^{K2060R}$). Mutation of this Lys residue results in the inability of ATPases, including the mammalian Kis homolog, CHD8 [71], to hydrolyze ATP and remodel chromatin [72, 73]. We expressed $UAS-kis^{K2060R}$ in all tissues, neurons, or postsynaptic muscles in $kis^{LM27}$/$kis^{k13416}$ mutants. We verified $Kis^{K2060R}$-V5 was expressed in the nuclei of each tissue (S5B Fig). Similar as CHD7-V5, we noted that there was some extranuclear/cytoplasmic signal, which was most apparent in postsynaptic tissue where $Kis^{K2060R}$-V5 exhibited a punctate nuclear and extranuclear distribution. Notably, this signal was not present in the muscle nuclei of larvae expressing $UAS-kis^{K2060R}$ in neurons. Expression of $Kis^{K2060R}$ in all tissues, neurons, or postsynaptic muscles in $kis^{LM27}$/$kis^{k13416}$ mutants restored both endocytosis (Fig 9C and 9D) and locomotor behavior (Fig 9A and 9B) as there were no significant differences in locomotion or endocytosis

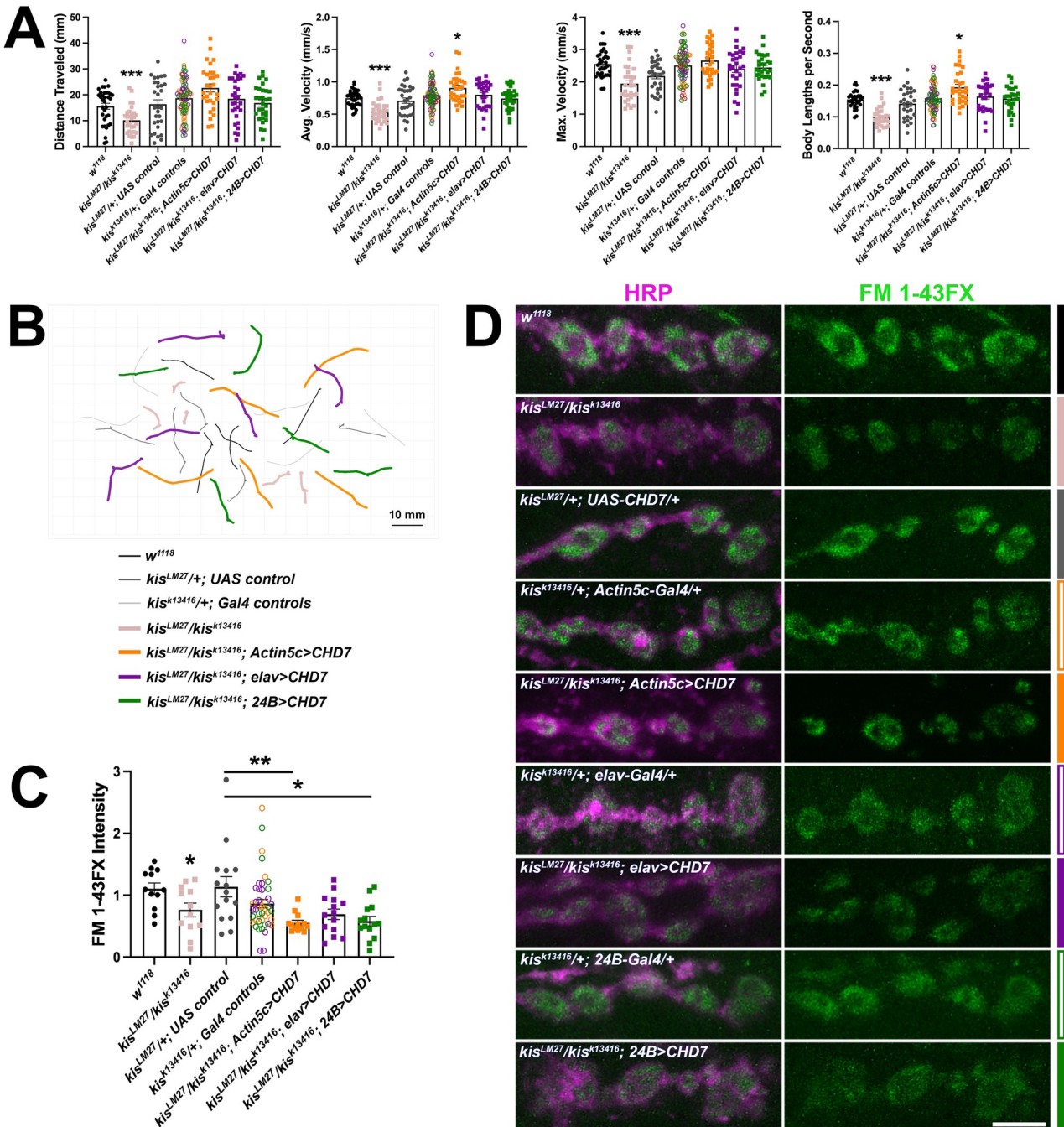

**Fig 8. Expression of human CHD7 in *kis* mutants restores behavior but not endocytosis.** Tissue-specific drivers were used to express human CHD7 in all tissues (using the *Actin5c-Gal4* driver), presynaptic motor neurons (using the *elav-Gal4* driver), or postsynaptic muscle (using the *24B-Gal4* driver) of *kis* mutants. A) Histograms show quantification as determined by wrMTrck of larval crawling behavior on an agar arena for 30 s. Distance traveled, maximum velocity, average velocity, and crawling velocity normalized to body size (body lengths per second). B) Representative traces of larval crawling behavior for the genotypes listed. C) Quantification of FM 1-43FX fluorescence intensity indicates that expression of human CHD7 does not restore endocytosis in *kis* mutants. D) High resolution confocal micrographs of terminal presynaptic motor neuron boutons (HRP, magenta) after internalization of the lipophilic dye FM 1-43FX (green) after 1 min stimulation with 90 mM KCl in genotypes as listed. Scale bar = 5 μm.

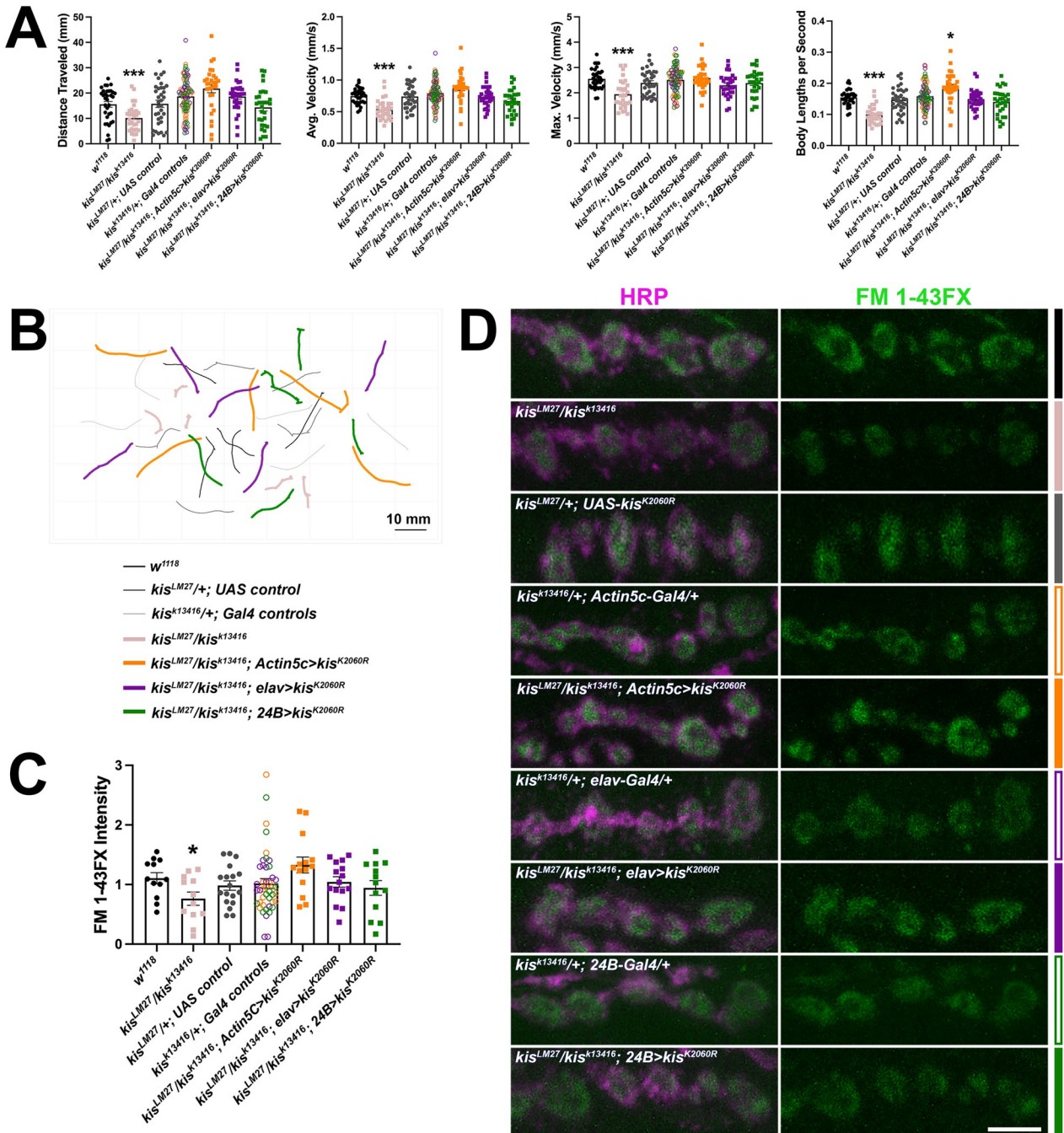

**Fig 9. Expression of an ATPase deficient Kis, Kis^K2060R, in *kis* mutants restores behavior and endocytosis.** Tissue-specific drivers were used to express Kis^K2060R in all tissues (using the *Actin5c-Gal4* driver), presynaptic motor neurons (using the *elav-Gal4* driver), or postsynaptic muscle (using the *24B-Gal4* driver) of *kis* mutants. A) Histograms show quantification as determined by wrMTrck of larval crawling behavior on an agar arena for 30 s. Distance traveled, maximum velocity, average velocity, and crawling velocity normalized to body size (body lengths per second). B) Representative traces of larval crawling behavior for the genotypes listed. C) Quantification of FM 1-43FX fluorescence intensity indicates that expression of Kis^K2060R restores endocytosis in *kis* mutants. D) High resolution confocal micrographs of terminal presynaptic motor neuron boutons (HRP, magenta) after internalization of the lipophilic dye FM 1-43FX (green) after one min stimulation with 90 mM KCl in genotypes as listed. Scale bar = 5 μm.

between experimental animals and outcrossed controls. These findings suggest that Kis may have functional implications at the synapse that are, at least partly, independent of its chromatin remodeling activity.

## Discussion

Our data indicate that Kis promotes both CME and ADBE possibly by regulating the expression of gene products that mediate these processes (Fig 1) and maintaining synaptic vesicle pools (Fig 6). Kis is important in postsynaptic muscles for endocytosis as postsynaptic Kis-L expression in *kis* mutants restores endocytosis (Fig 7). Surprisingly, the ATPase activity of Kis is dispensable for endocytosis and locomotion (Fig 9).

### Kis promotes both CME and ADBE

Several forms of endocytosis exist in neurons to collectively internalize nutrients, detect growth and guidance cues, and maintain synaptic membrane homeostasis [13]. While several endocytic mechanisms are thought to replenish synaptic vesicles [11], there is little evidence for kiss-and-run at the *Drosophila* NMJ [74]. Compensatory endocytosis and ultrafast endocytosis are thought to occur at the NMJ but there are few direct demonstrations of these mechanisms [12]. CME is thought to be the most prevalent form of endocytosis at the synapse [11] but is limited by the duration required to select cargo and activate CME proteins [75]. Other modes of endocytosis, therefore, compensate for these inherent limitations in CME to enable synaptic homeostasis. ADBE, which quickly internalizes large areas of membrane, is utilized during periods of intense neuronal activity [14]. Kis functionally contributes to both forms of endocytosis as evidenced by the reduction in *kis* mutant eEJC amplitudes during stimulation protocols that induce ADBE (Fig 2) or CME (Fig 3). Further, when ADBE or CME are pharmacologically inhibited, *kis* mutants do not show further reductions in endocytosis (Figs 4 and 5).

Postsynaptic Kis-L expression in *kis* mutants rescues endocytosis (Fig 7A and 7B). Similarly, knock down of Kis in postsynaptic muscle and all tissues mimics the $kis^{LM27}/kis^{k13416}$ mutant phenotype (Fig 7C). These data indicate that postsynaptic Kis regulates presynaptic endocytosis. Although this may seem unexpected, retrograde signaling via cell adhesion molecules (CAMs) and diffusible molecules released postsynaptically regulate many presynaptic processes including the synaptic vesicle cycle [76]. Presynaptic endocytosis is enhanced by production of nitric oxide downstream of N-methyl D-aspartate (NMDA) receptor signaling, which increases production of the membrane lipid phosphatidylinositol 4,5-biphosphate (PIP$_2$) [77]. PIP$_2$ recruits adaptor protein 2 (AP2) to the plasma membrane [78] to enable endocytosis by selecting cargo and binding to clathrin [79]. Transsynaptic crosstalk through CAMs also modulates endocytosis as null mutations in Neuroligin 1 (Nlg1) enhance endocytosis without affecting exocytosis or the size of vesicle pools [80].

Kis restricts the synaptic localization of the CAM FasII [30], the *Drosophila* ortholog of neural cell adhesion molecule (NCAM). Similarly, loss of function of *Chd7* [81] or *Chd8* [82] results in differential expression of several CAMs including *ncams* and *nlgs* as determined by ChIP-Seq. Kis binds near the regulatory regions of *fasII*, *fasIII*, *nlg2*, and *nlg4* in *Drosophila* intestinal stem cells [83]. NCAM negatively regulates ADBE [84], possibly by activating protein kinase B/Akt [85, 86], and binds directly to AP2 [87]. Therefore, mutations in *kis* may lead to deficient endocytosis due to the accumulation of CAMs, which might physically restrict invagination of the plasma membrane.

Alternatively, Kis may regulate CME and ADBE by regulating proteins that organize microdomains of the synapse like Rabs. Rab GTPases facilitate directional membrane trafficking between compartments of the endomembrane system [88]. Specifically, Rab11 traffics cargo

between recycling endosomes and the plasma membrane [89] to influence the concentration of lipids and membrane-associated proteins [90]. Kis promotes the transcription of *rab11* and Rab11 localization to the synapse [31]. Rab11, when constitutively active, increases the number of rat cerebellar granule cell neuron terminals executing ADBE. Nlg2, Nlg3, and NCAM were present in bulk endosomes [91] suggesting that Rab11 controls the localization of CAMs by regulating ADBE. Thus, Kis may functionally influence endocytosis by a combination of mechanisms including, but not limited to, regulating the expression of endocytic gene transcripts, expression and/or localization of endocytic proteins, CAMs, and Rab11.

Kis is 63% identical to human CHD7 [26]. Despite this conservation, expression of human CHD7 in all tissues or postsynaptic muscles of *kis* mutants did not restore endocytosis (Fig 8C and 8D) as was observed with *UAS-kis-L* expression in the same tissue types (Fig 7A and 7B). CHD family chromatin remodeling enzymes assemble with DNA binding proteins [92], histone methytransferases [93], and other proteins to form large, multisubunit complexes. Although the organization and identities of the subunits and the mechanism by which chromatin remodeling enzyme complexes recognize nucleosomes is poorly understood [94], it is recognized that the complexes are developmental- and tissue-specific [95, 96]. Therefore, it is possible that overexpression of CHD7 might produce a dominant negative effect in *kis* mutants where CHD7 sequesters other transcriptional regulators rendering them inactive. This possibility, however, seems unlikely because CHD7 expression in all tissues, neurons, or postsynaptic muscle of *kis* mutants restored locomotor behaviors (Fig 8A and 8B). It is more plausible that this discrepancy arises because different tissue types are required for locomotion and endocytosis. Locomotor behaviors require broader, multi-circuit coordination compared with endocytosis. Larval locomotion is produced by central nervous system central pattern generators, which are subject to upstream cholinergic control, and peripheral motor neurons. This circuit may be initiated independent of, but is modulated by, sensory feedback [97]. Presynaptic endocytosis is influenced by interactions between pre- and postsynaptic cells and associated glia [98]. CHD7 binds to unique regulatory loci in different tissues including human neural crest cells, induced pluripotent stem cell-derived neuroepithelial cells, and induced pluripotent stem cells. Further, CHD7 preferentially associates with the transcription factors TFAP2A and NR2F1/F2 in neural crest cells but not neuroepithelial cells [99]. Thus, more CHD7 sensitive gene products may be required for locomotion compared with endocytosis.

## Kis may transcriptionally regulate endocytosis

CHD7, CHD8, and Kis influence the transcription of thousands of gene products [81, 83, 100, 101]. Therefore, it is not surprising that Kis affects the transcript levels of several endocytic genes including *AP2α*, *endoB* [31], *PI3K92E*, and *sgg/GSK3β* (Fig 1). In *Drosophila* adult intestinal stem cells, Kis was bound near regulatory sequences for several endocytic genes including *amphiphysin*, *AP2α*, *clathrin light* and *clathrin heavy chain*, *dyn*, *draper 1*, *flower*, *lqf*, and *synaptojanin* [83]. We did not observe altered transcript levels of *clathrin heavy chain* (Fig 1), *amphiphysin*, *flower*, *lqf*, or *synaptojanin* [31] via RT-qPCR. Similarly, we did not detect Kis binding to the promoter or within 200 bp of the transcription start site of *dyn* even though Kis is enriched at the promoters and transcription start sites for *dap160* and *endoB* [31]. These discrepancies likely reflect developmental- and/or tissue-specific differences in chromatin remodeling complexes as a result of differential expression of specific transcription factors [96, 102].

Kis, CHD7, and CHD8 contain several domains including two chromodomains, which bind to methylated N-terminal histone residues [103, 104], and an ATPase domain [105]. The latter repositions histones relative to DNA to expose transcriptional regulatory sequences [106]. We

found that expression of an ATPase-dead Kis, Kis[K2060R], in *kis* mutants restored locomotor behaviors and endocytosis (Fig 9). Similarly, expression of Kis[K2060R] partially rescued the size of stem cell clones in the *kis[10D26]* loss of function mutant [83]. Taken together, these data suggest that Kis may function independent of its ATPase domain to form functional regulatory transcriptional complexes. Chromatin remodeling proteins are integral components of large, multi-subunit complexes, the composition of which differ in a cell- and developmental-dependent manner [95, 96]. In *Drosophila* intestinal stem cells, Kis colocalizes with the histone methyltransferase, Trithorax related (Trr), to regulate the transcription of target genes [83]. Kis recruits the histone methyltransferases, Trithorax and Absent, Small or Homeotic Discs 1 (ASH1) to *Drosophila* salivary polytene chromosomes [93]. Similarly, CHD8 recruits ASH2L, which is part of a lysine methyltransferase complex, to regulate gene expression required for differentiation of mouse oligodendrocyte precursor cells. Notably, differentiation was partially restored by inhibition of KDM5, a demethyltransferase, in conditional *Chd8* knock outs [107] indicating that CHD8's role in oligodendrocyte precursor cell differentiation occurs partly because of its ability to recruit histone methyltransferases. KDM5 interacts with Kis and Trr in *Drosophila* adult heads [108] indicating these complexes may be similar across species.

Collectively, these data are consistent with clinical findings, which show that mutations throughout CHD proteins perturb their function. Approximately 30% of CHARGE-associated *Chd7* mutations lie within functional domains including the ATPase domain [28] and the percentage of *Chd8* point mutations within the ATPase domain associated with autism spectrum disorders is 8% [109]. Thus, residues throughout CHD7 and CHD8 contribute to their transcriptional function. Indeed, *Chd7* mutations in CHARGE patients produce different phenotypes depending on the location with mutations in the chromodomains resulting in the most severe phenotypes [28]. Similarly, while mutations in the chromodomains of zebrafish *chd7* produce the morphological characteristics of CHARGE Syndrome including craniofacial defects, mutations in the ATPase domain do not [110].

Alternatively, the capacity of Kis[K2060R] to restore locomotor behaviors and endocytosis in *kis* mutants may occur due to a novel cytoplasmic role for Kis. Other components of chromatin remodeling complexes including the MYST family of acetyltransferases [111] and members of the histone deacetyltransferase (HDAC) family [112] are localized in both the nucleus and cytoplasm where they recognize cytoplasmic substrates. CHD1 [113] and CHD9 [114] were detected in the cytoplasm of mitotic cells. We have not detected Kis in the cytoplasm of ventral nerve cord neurons or postsynaptic muscle cells [30]. Given that 1) Kis binds to regulatory regions of the endocytic genes *dap160* and *endoB* [31], 2) chromatin remodeling enzymes are integral components of transcriptional complexes that include additional proteins that influence transcription, and 3) mutations outside the ATPase domain of CHD proteins exert the most severe phenotypes, we favor the hypothesis that Kis may influence transcription independent of its ATPase domain.

## Implications for neurodevelopmental disorders and neurodegenerative diseases

The synaptic vesicle cycle, which includes processes that enable release and recycling of vesicles, is disrupted in neurodevelopmental disorders [115] and neurodegenerative diseases [76]. Genes that regulate the synaptic vesicle cycle are differentially expressed in human Alzheimer's disease patient neurons [116–118]. It is unclear, however, whether these genetic changes precede the formation of amyloid β (Aβ) plaques, which are produced by proteolytic cleavage of amyloid precursor protein (APP) [25]. Cellular exposure to Aβ40 and Aβ42 aggregates impairs endocytosis in PC12 and neuroblastoma cells and Aβ40 slows the intracellular trafficking of

recycling endosomes [116]. Impaired trafficking of endosomes containing amyloid precursor protein (APP) exacerbates production of Aβ40 and Aβ42 as the amyloidogenic pathway of APP processing is favored [119].

Although it is recognized that chromatin remodeling is a contributing factor to neurodevelopmental disorders including autism spectrum disorders [120], how aberrant chromatin remodeling functionally influences the synaptic vesicle cycle is relatively unexplored. We find that Kis maintains synaptic vesicle pools as *kis* mutants exhibit reductions in the sizes of the readily releasable, recycling, and reserve pools of vesicles (Fig 6). CME and ADBE both replenish synaptic vesicle pools [12, 13] but the extent to which each contributes is not well understood. Because the synaptic vesicle cycle is intimately linked to the endomembrane system [121], it may be difficult to distinguish discrete pools of vesicles. The readily releasable pool of vesicles was regenerated from CME while the reserve pool was bolstered by ADBE in cultured rat cerebellar neurons [122]. CME replenished 63% of the vesicle pool in cultured hippocampal neurons at 9 DIV but only 39% at 19 DIV indicating ADBE is preferentially used in mature neurons to replenish synaptic vesicles [123].

The reduced size of synaptic vesicle pools in *kis* mutants may lead to the decrease in evoked potentials [30], failed homeostatic presynaptic plasticity [124], and impaired motor behavior (Figs 7D, 8A and 8B, 9A and 9B) observed in *kis* mutants. It is not clear whether deficient endocytosis in *kis* mutants leads to the decrease in the synaptic vesicle pool or whether *kis* mutants produce fewer synaptic vesicles de novo. Our data begins to uncover correlates between aberrant chromatin remodeling and synaptic processes. Given the altered state of chromatin in neurodevelopmental disorders and neurodegenerative diseases [125], a better understanding of the synaptic correlates affecting cognitive and behavioral processes is needed.

## Supporting information

**S1 Fig. CME is inhibited by Dynasore but not BAPTA or EGTA.** eEJCs were recorded during 5 Hz stimulation for five minutes in HL-3 + 1.0 mM $Ca^{2+}$. 100 μM BAPTA (n = 10), 25 μM EGTA (n = 10), 100 μM Dynasore (n = 9), or an equal volume of DMSO (controls, n = 11) were applied for 10 minutes prior to neuronal stimulation. Each eEJC is normalized to the first stimulus for each condition. Points represent mean relative eEJC amplitudes. Error bars represent the SEM.
(TIF)

**S2 Fig. ADBE is not inhibited by BAPTA, EGTA, or Dynasore.** eEJCs were measured in HL-3 + 1.0 mM $Ca^{2+}$ for 60 sec of 20 Hz HFS to induce ADBE and during a 50 sec recovery period with 0.2 Hz stimulation. 100 μM BAPTA (n = 10), 25 μM EGTA (n = 9), 100 μM Dynasore (n = 9), or an equal volume of DMSO (controls, n = 10) were applied for 10 minutes prior to neuronal stimulation. Each eEJC is normalized to the first stimulus for each condition. Points represent mean relative eEJC amplitudes. Error bars represent the SEM.
(TIF)

**S3 Fig. Genotype by genotype comparison of ADBE and CME compounds.** Endocytosis was assessed by measuring internalization of the lipophilic dye FM 1-43FX after one min stimulation with 90 mM KCl. Panels show high resolution confocal micrographs of terminal presynaptic motor neuron boutons (HRP, magenta) after internalization of the lipophilic dye FM 1-43FX (green). Data for each genotype were normalized to the DMSO control condition.
(TIF)

**S4 Fig. Restoration of Kis in all tissues rescues endocytosis.** The *Actin5c-Gal4* driver was used to express *UAS-kis-L* in all tissues of *kis* mutants. A) High resolution confocal micrographs of terminal presynaptic motor neuron boutons (HRP, magenta) after internalization of the lipophilic dye FM 1-43FX (green) after one min stimulation with 90 mM KCl in genotypes as listed. Scale bar = 5 μm. B) Quantification of FM 1-43FX fluorescence. C) Relative expression of CNS transcripts was assessed via RT-qPCR. $2^{-\Delta\Delta C(t)}$ values are indicated. Data includes four biological replicates each including three technical replicates. Technical replicates are represented by the points for the representative genotypes. Bars indicate the SEM.
(TIF)

**S5 Fig. CHD7-V5 and Kis$^{K2060R}$-V5 are localized to the nuclei of tissues.** Confocal micrographs showing V5 (green) and HRP (magenta, neuron) immunolabeling and DAPI labeling. Left and middle large panels show representative ventral nerve cords with single nuclei depicted in the small panels of *kis* mutants expressing human CHD7 (A) or the ATPase deficient Kis$^{K2060R}$ (B). Scale bar = 5 μm. Right panels show representative muscles with single nuclei depicted in the small panels of *kis* mutants expressing human CHD7 (A) or the ATPase deficient Kis$^{K2060R}$ (B). Scale bar = 5 μm.
(TIF)

**S1 Table. Statistical comparisons and corresponding p-values for figures.**
(XLSX)

## Acknowledgments

We thank the Bloomington *Drosophila* Stock Center for fly stocks (NIH P40OD018537), Carly Gridley for help with the FM 1-43FX rescue experiments, Dave Featherstone for his mentoring, and Southern Illinois University Edwardsville for travel support.

## Author Contributions

**Conceptualization:** Faith L. W. Liebl.

**Data curation:** Faith L. W. Liebl.

**Formal analysis:** Emily L. Hendricks, Faith L. W. Liebl.

**Funding acquisition:** Faith L. W. Liebl.

**Investigation:** Emily L. Hendricks, Faith L. W. Liebl.

**Methodology:** Emily L. Hendricks, Faith L. W. Liebl.

**Project administration:** Faith L. W. Liebl.

**Resources:** Faith L. W. Liebl.

**Software:** Emily L. Hendricks.

**Supervision:** Faith L. W. Liebl.

**Validation:** Faith L. W. Liebl.

**Visualization:** Emily L. Hendricks, Faith L. W. Liebl.

**Writing – original draft:** Emily L. Hendricks, Faith L. W. Liebl.

**Writing – review & editing:** Emily L. Hendricks, Faith L. W. Liebl.

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
