## [Decision Letter · Decision Letter 0]

14 Sep 2023

PONE-D-23-22620The CHD family chromatin remodeling enzyme, Kismet, promotes both clathrin-mediated and activity-dependent bulk endocytosisPLOS ONE

Dear Dr. Liebl,

Thank you for submitting your manuscript to PLOS ONE. After careful consideration, we feel that it has merit but does not fully meet PLOS ONE’s publication criteria as it currently stands. Therefore, we invite you to submit a revised version of the manuscript that addresses the points raised during the review process.

We look forward to receiving your revised manuscript.

Kind regards,

Alexander G Obukhov, Ph.D.

Academic Editor

PLOS ONE

Journal Requirements:

"We thank the Bloomington Drosophila Stock Center for fly stocks (NIH P40OD018537), Carly Gridley for help with the FM 1-43FX rescue experiments, Dave Featherstone for his mentoring, and Southern Illinois University Edwardsville for travel support. This work was funded by the National Institute of Neurological Disorders and Stroke of the NIH under award numbers 1R15NS101608-01A1 and 2R15NS101608-02A1 (to FL) and by Southern Illinois University Edwardsville’s Competitive Graduate Award (to EH). The funders had no role in study design, data collection and analysis, decision to publish, or preparation of the manuscript."

"This work was funded by the National Institute of Neurological Disorders and Stroke of the NIH under award numbers 1R15NS101608-01A1 and 2R15NS101608-02A1 (to FL) and by Southern Illinois University Edwardsville’s Competitive Graduate Award (to EH). The funders had no role in study design, data collection and analysis, decision to publish, or preparation of the manuscript."

4. Please include complete captions for your Supporting Information files at the end of your manuscript, and update any in-text citations to match accordingly. Please see our Supporting Information guidelines for more information: http://journals.plos.org/plosone/s/supporting-information. 

Reviewers' comments:

Reviewer's Responses to Questions

**Comments to the Author**

1. Is the manuscript technically sound, and do the data support the conclusions?

Reviewer #1: Yes

Reviewer #2: Partly

Reviewer #3: Partly

2. Has the statistical analysis been performed appropriately and rigorously? 

Reviewer #1: No

Reviewer #2: No

Reviewer #3: No

3. Have the authors made all data underlying the findings in their manuscript fully available?

Reviewer #1: Yes

Reviewer #2: No

Reviewer #3: Yes

4. Is the manuscript presented in an intelligible fashion and written in standard English?

Reviewer #1: Yes

Reviewer #2: Yes

Reviewer #3: Yes

5. Review Comments to the Author

Reviewer #1: In this work, the authors investigate the role of the chromatin remodeling enzyme Kismet in synaptic endocytosis at Drosophila neuromuscular junctions, an excellent and popular system for studying molecular mechanisms of synaptic vesicle endocytosis. The work builds on previous publications by the authors showing that kismet is important for recycling synaptic vesicles during endocytosis.

In extensive experiments using kis mutants and rescue and RNAi approaches, they show that Kismet promotes clathrin-mediated and activity-dependent mass endocytosis of synaptic vesicles. They also show that kismet is important in postsynaptic muscle to promote presynaptic endocytosis as well as locomotion. Interestingly, this function seems to be independent of the ATPase domain.

Although the molecular mechanism of how kismet mediates these processes is still unclear, this work represents an important and novel contribution to the study of the physiological function of kismet. The experiments seem to be carefully executed, however, some points, especially related to evaluation of experiments and statistical analysis, need to be addressed in more detail.

Major points:

1) Experimental details, sample size: Throughout the manuscript, there is no indication of how many animals/samples were used for each experiment. It is only stated in general terms that at least two independent biological experiments were performed. In addition, it is not clear what the sample points shown present. For example, in the endocytosis experiments, does a sample point represent the analysis of a section or a projected z-stack slice? How many sections per larva and how many larvae were used? Electrophysiology - how many experiments were performed with how many larvae? Is the mean value given? What about error bars? All this is of course important for the later statistical analysis and must therefore be indicated in the legends to the figures of the individual experiments.

2) Statistics: The authors state that they used an unpaired t-test when there was only one control. This is not correct, because for experiments with more than two groups, the comparison must be made with a one-way ANOVA and an appropriate post-hoc test. This is independent of the number of control groups. Using an ANOVA you can be confident that any statistically significant result you find is not just due to running many tests. In addition, authors should check for normality distribution.

3) Quantification of endocytosis: as I understand it, the authors measured FM 1-43X intensity. Can the authors rule out an effect of the genetic manipulations/treatments on the size/area of the boutons and thus on the fluorescence intensity of FM 1-43X? Have they also measured HRP fluorescence intensity and is it comparable between NMJs?

4) Figure 1: On what basis were the 10 genes selected to be analyzed by qPCR? Some explanation would be appreciated. While the results for sgg are convincing, three data points that raise the mean dominate the results for PI3K92E and synd, especially for the Kis13416 mutant. What does a data point mean in this context? Does one data point represent one animal/larva? And why is the number of data points different for each gene?

5) Figure S1: Why did BAPTA and EGTA not have an effect on CME or ABDE in control animals? Since they are supposed to inhibit CME and/or ABDE, the finding is surprising.

6) Figure 4 and 5: The presentation of the data (graphs) is very confusing. Here, the untreated and treated genotypes need to be compared to assess how strong the effect of the treatment is in WT and mutants and to judge if the treatments have any effect compared to the respective DMSO control.

7) Figure 7D: The authors conclude from the data shown that Kis functions in all tissues and neurons to promote proper local locomotion. However, reexpression of Kis in postsynaptic muscle increases larval movement while the reexpression in other tissues had no effect. How does that support the conclusion? Also, why is the control group KisLM27/Kisk13416 missing in this graph? It is very confusing to have to jump to the controls of Figures 8 and 9 first to understand Figure 7D.

8) Line 369: UAS-kis-L data should be shown.

9) Line 451: Please show the unpublished data as supplement

Reviewer #2: This paper extends prior work from the lab showing roles for CHD genes in endocytosis by examining the role of the Drosophila kismet in synaptic vesicle recycling. The authors report differences in both clathrin mediated endocytosis and a backup endocytic pathway, activity dependent bulk endocytosis and then use tissue-specific rescue and knock down experiments to try to tease out the where kismet is required for its functions, with results pointing to a role in post-synaptic muscles. Further ATPase- mutants of kismet rescues both endocytosis and movement dysfunction in the mutant whereas human CHD7 only rescues movement functions. Overall, the study adds to our understanding of this large protein and its novel role in synapses. The discussion is thought provoking and contextualizes the work.

Prior to publication, I have major issues that need to be addressed, most important of which is related to Figures 4 and 5 and the use of drugs and their interpretation:

• In figure 4, stars for significance are hard to distinguish from plotted data.

The quantification in the graphs does not appear to match the images shown, which present a much greater than 2-fold reduction in intensity. More representative average images should be shown. Furthermore it is unclear how the ratios are assessed here. They state that they are normalizing all of the samples to w1118 but the exposure cases should be normalize to the vehicle control to determine if the pre-treatment has an effect. In that case, it would appear (from the images provided) that kis is affected by Roscovitine but not EGTA or BAPTA, but dap is only affected by BAPTA and for Iqf and sgg it is unclear because the controls are already very weakly green. The interpretation that this shows kis affects just ADBE seems unjustified. I have similar concerns about the images in Figure 5 and the quantification.

More detail should be provided about the molecular nature of each inhibitor and what it does. It is unclear why different mutants would respond differently to inhibitors of the same process.

• Fig 2. BAPTA and Dynasore are used on the controls, but only the latter on the mutants to show involvement in CME. Both should be tested on the mutants to corroborate the the findings.

• In figure 2A, the curves for kis and Iqf practically overlap but it is shown that only kis mutants are statistically different than control. This is not at all apparent why this would be the case, especially since we do not see error bars for each time point. I think expanding the Y axis in the region from 50-85 would be helpful to see the differences that are being claimed.

Again in figure 7, it seems to me that the statistical differences should be tested between the kis heterozygote and homozygote with the gal4 tissue-specific driver not compared with the UAS-kis-L pan driver. It is also unclear why the kis/+; 24B-GAL4/+ has reduced endocytosis in the first place. It suggests that there is a defect in endocytosis caused by the Gal4 drives itself.

Minor details that should be addressed:

• It should be stated that sgg is on the X chromosome necessitating the use of females since non-fly readers do not know that the first chromosome is the X.

• Significance is not marked in Figure 1. Are any significantly different from controls?

• It is stated that “Recovery from HFS was impaired in kis mutants from 40-50 seconds after HFS 240 and at 40 seconds after HFS in lqf mutants,” but the significance starts show differences at 25s and 30s and respectively.

Line 358, “Similary” should be “Similarly”

Line 426 “for both endocytosis”. Only one item is listed. Remove “both”

Line 437. I think you mean Figs 4,5 not 5,6

Line 480-481: “This circuit may be initiated independent of, but is modulated by, sensory feedback”. needs punctation as marked

Are scale bars the same for all images in Figs 4,5, 7? Were these slides imaged with identical settings?

Reviewer #3: In this study, Hendricks and Liebl assess roles for Drosophila Kismet, a chromodomain family (CHD) protein, in regulating presynaptic endocytosis. Alleles of the mammalian homologs of Kis (CHD7 and CHD8) are linked to autism spectrum disorders and CHARGE syndrome, and are thought to regulate a variety of cell functions including cell adhesion and endocytosis, possibly through chromatin remodeling and/or transcriptional regulation. Using reduction-of-function alleles, the authors report that Kis participates in clathrin-mediated and clathrin-independent endocytosis. Surprisingly, its expression in postsynaptic cells appears to be responsible for its role in presynaptic endocytosis, although the mechanisms are currently unclear.

The study itself presents novel and interesting observations for roles of Kis in endocytosis and, even though they do not yet hint at the mechanism of action, this work should be a relevant beginning point for future work examining relationships between CHD proteins, endocytosis and/or regulation of the synaptic vesicle pool, and neuronal function. In places, the authors report counterintuitive findings but do not provide explanations, and as a result many aspects of this manuscript were difficult to follow. The manner in which data were presented in the figures also created challenges.

Specific comments:

1. Many of graphs were very difficult to interpret when printed at page size. Making the graphs larger or increasing font size of axis labels would be helpful.

2. Throughout the manuscript, the authors distinguish between two forms of endocytosis: clathrin-mediated (CME) and activity-dependent bulk endocytosis (ADBE, which is clathrin-independent). Even in the abstract, the authors state that endocytosis occurs via these two mechanisms, but this is an oversimplification as there are numerous clathrin-independent endocytic pathway. At neurons, ultrafast endocytosis is a clathrin-independent pathway that is distinct from ADBE and which appears to play a prominent role in replenishment of the SV pool. Even if the focus of this study is on CME versus ADBE, the authors should broaden their discussion of endocytic pathways throughout the manuscript to include the likelihood that other pathways are also involved.

3. In Figure 1, the authors describe changes in expression for genes linked to CME, ADBE, or both using a kis hypomorph (k13416) and a loss-of-function (heterozygous LM27/k13416). They report 50% reductions of 150% increases in expression for some genes compared to WT, but the significance (statistical and biological, especially since transcriptionally inactive Kis remains functional for endocytosis and locomotion in their later analyses) is not clear. Statistical analyses should be performed to compare the LM27/k13416 to WT controls and to the k13416 hypomorph.

4. The curves in Figs. 2A and 3A, with accompanying explanation, were confusing. In the text (lines 239-40), the authors state that “recovery from HFS was impaired in kis mutants from 40-50 seconds after HFS and at 40 seconds after HFS in lqf mutants”. Later (lines 253-5), the authors state that kis mutants exhibit a more severe reduction than lqf mutants. In both cases, the wording in the text implies differences between the two mutants, but curves for kis and lqf look essentially identical to one another. At the least, the wording is imprecise and should be modified, but as it stands the data do not support the authors’ conclusions. The curves (as well as those in Figs. 2C, S1 and S2) lack error bars that would add context. Related to this, the fact that Lqf (epsin) is directly involved in CME as an adaptor, and its dephosphorylation triggers ADBE, suggests that it plays roles in both processes. As a result, it is difficult to interpret data using lqf mutants in the context of separating roles in CME versus ADBE.

5. The authors use Dynasore to inhibit dynamin, with the aim of blocking CME. In Figures S1 and S2, they report that Dynasore inhibited CME but not ADBE. This is surprising because there is a solid body of evidence in the literature demonstrating that dynamin is involved in both endocytic pathways [e.g., Winther et al. (2013) J Cell Sci 126:1021-31 and others]. The reported effect on CME (Fig. S1) is surprisingly small and not consistent across time. Without error bars on these plots it is really difficult to assess whether there is or is not a difference, and the small magnitude of change is not convincing.

6. The finding that postsynaptic Kis is required for presynaptic endocytosis is interesting, but counterintuitive. The authors speculate extensively about possible explanations, but do not offer any experimental evidence to support any of these explanations. As a result, these results are descriptive. In the final paragraph of the discussion (lines 533-4), the authors state that “our data begins to uncover potential mechanisms by which aberrant chromatin remodeling affect synaptic processes,” but this study is not mechanistic in nature and this statement should be re-phrased.

7. The methods do not adequately describe how FM1-43 uptake was quantified. Specifically, did the authors correct for size of the regions they measured?

8. The graphs for Fig. 4 are missing labels to identify the samples with blue data points.

9. Wording of the text accompanying Fig. 7 was extremely confusing, and the experiments appear to lack some controls. This section may be difficult to follow for someone unfamiliar with Drosophila genetics. The data shown in Fig. 7 (images and graphs) needs to include WT and kis controls as a frame of reference for the re-expression experiments; moreover, a Kis-restored UAS should be included to show results for whole-organism re-expression. Statistical analyses should be performed relative to WT or kis mutant controls, in addition to the driver-specific controls. Why do the authors see significance compared to the UAS control but not compared to the driver-specific control in Fig. 7B?

10. On lines 357-8, the authors describe Kis knockdown from a previous study in wing discs, but they should also include confirmation and quantification of knockdown in their experimental setup.

11. The finding that transcriptionally-inactive Kis restores endocytosis and locomotor behavior (Fig. 9) is surprising and interesting. Immediately after presenting these results, the first paragraph of the discussion states that “our data indicate that Kis promotes both CME and ADBE, possibly by regulating the expression of gene products that mediate these processes” (lines 424-5). It is true that the authors report changes in gene expression in Fig. 1 (although statistical analyses were not presented as noted above), but this statement disagrees with the data from Fig. 9, which suggest that Kis acts independently of its role in regulating gene expression. There is an entire section of the discussion (lines 487-508) devoted to the possible role of Kis in transcriptional regulation of endocytosis, but this may not be relevant given the authors’ findings.

6. PLOS authors have the option to publish the peer review history of their article (what does this mean?). If published, this will include your full peer review and any attached files.

Reviewer #1: No

Reviewer #2: No

Reviewer #3: No

---

## [Author Response · Author response to Decision Letter 0]

25 Oct 2023

We thank the reviewers for the time they dedicated to carefully reviewing our manuscript. We appreciate their insightful comments and are confident that their feedback has improved the manuscript. Reviewer comments are shown in normal text. Our responses are in blue.

Reviewer #1: In this work, the authors investigate the role of the chromatin remodeling enzyme Kismet in synaptic endocytosis at Drosophila neuromuscular junctions, an excellent and popular system for studying molecular mechanisms of synaptic vesicle endocytosis. The work builds on previous publications by the authors showing that kismet is important for recycling synaptic vesicles during endocytosis.

In extensive experiments using kis mutants and rescue and RNAi approaches, they show that Kismet promotes clathrin-mediated and activity-dependent mass endocytosis of synaptic vesicles. They also show that kismet is important in postsynaptic muscle to promote presynaptic endocytosis as well as locomotion. Interestingly, this function seems to be independent of the ATPase domain.

Although the molecular mechanism of how kismet mediates these processes is still unclear, this work represents an important and novel contribution to the study of the physiological function of kismet. The experiments seem to be carefully executed, however, some points, especially related to evaluation of experiments and statistical analysis, need to be addressed in more detail.

Major points:

1) Experimental details, sample size: Throughout the manuscript, there is no indication of how many animals/samples were used for each experiment. It is only stated in general terms that at least two independent biological experiments were performed. In addition, it is not clear what the sample points shown present. For example, in the endocytosis experiments, does a sample point represent the analysis of a section or a projected z-stack slice? How many sections per larva and how many larvae were used? Electrophysiology - how many experiments were performed with how many larvae? Is the mean value given? What about error bars? All this is of course important for the later statistical analysis and must therefore be indicated in the legends to the figures of the individual experiments.

The sample points shown represent the sample size. This information was included in the methods: 

Bar graphs in figures show sample sizes. Statistical significance is represented on bar graphs as follows: * = <0.05, ** = <0.01, *** = <0.001 with error bars representing standard error of the mean (SEM). Table S1 shows the statistically significant p - values corresponding to each figure.

However, we have amended the text in the methods to include that sample points for endocytosis experiments represent the mean of the z-stack slices and means are always shown in the graphs. We have also added this information to the figure legends for the figures that show electrophysiological data. For all experiments, the number of Z-sections vary from larva to larva as the depth of the NMJ varies from animal to animal. There was also a supplemental table included to show p-values for all experiments.

2) Statistics: The authors state that they used an unpaired t-test when there was only one control. This is not correct, because for experiments with more than two groups, the comparison must be made with a one-way ANOVA and an appropriate post-hoc test. This is independent of the number of control groups. Using an ANOVA you can be confident that any statistically significant result you find is not just due to running many tests. In addition, authors should check for normality distribution.

Our statistics were performed as the reviewer described. Unpaired t-tests were used, for example, to compare the kis mutant to its w1118 control. For any experiment that included more than one control, and therefore more than two groups, one-way ANOVAs were performed with post hoc analyses. Post hoc tests utilized were determined by the “normality distribution”, which was indicated by Bartlett’s test for homogeneity of variances. Because the reviewer raised this concern, we amended the “Experimental Design and Statistical Analyses” section of the methods to clarify this. 

3) Quantification of endocytosis: as I understand it, the authors measured FM 1-43X intensity. Can the authors rule out an effect of the genetic manipulations/treatments on the size/area of the boutons and thus on the fluorescence intensity of FM 1-43X? Have they also measured HRP fluorescence intensity and is it comparable between NMJs?

Although there are not published differences in bouton sizes for any of the genotypes used in our experiments compared to controls, we understand the reviewer’s concern. Fluorescence intensity measurements in Fiji provide the mean pixel intensity normalized for the region of interest/area. Therefore, fluorescence intensity values are not influenced by potential size differences between NMJs. We have not measured HRP fluorescence intensity. Our measurements of FM 1-43FX signal intensity are consistent with other publications 1-3. 

4) Figure 1: On what basis were the 10 genes selected to be analyzed by qPCR? Some explanation would be appreciated. While the results for sgg are convincing, three data points that raise the mean dominate the results for PI3K92E and synd, especially for the Kis13416 mutant. What does a data point mean in this context? Does one data point represent one animal/larva? And why is the number of data points different for each gene?

The gene products were selected because of their roles in ADBE and/or CME as described in the results. The data points represent technical replicates and this information has been added to the figure legend. There are 9-12 data points for each gene product. The number of data points/technical replicates vary based on the number of biological replicates (3-4) and whether individual reactions produced product.

5) Figure S1: Why did BAPTA and EGTA not have an effect on CME or ABDE in control animals? Since they are supposed to inhibit CME and/or ABDE, the finding is surprising.

We were also surprised by these results. While BAPTA showed decreased eEJC amplitudes during both low and high frequency stimulation, these reductions were not statistically significant. We followed published protocols for the duration of exposure and these are referenced in the text.

6) Figure 4 and 5: The presentation of the data (graphs) is very confusing. Here, the untreated and treated genotypes need to be compared to assess how strong the effect of the treatment is in WT and mutants and to judge if the treatments have any effect compared to the respective DMSO control.

We agree that there is more than one way to present these data given that both the compounds and genotypes represent the independent variables for these experiments. The experiments were performed one compound at a time instead of one genotype at a time. Therefore, the graphs shown in Figures 4 and 5 quantify mean relative FM 1-43FX intensities for each compound used. The images shown and data obtained for these experiments were performed on the same days. Comparisons of compounds across genotypes were performed on separate days but, given that the same number of controls were used each day, we created a separate set of graphs to compare the results on a compound by compound basis and these data are shown in Fig S3. These data show that all compounds impaired endocytosis in w1118 controls.

7) Figure 7D: The authors conclude from the data shown that Kis functions in all tissues and neurons to promote proper local locomotion. However, reexpression of Kis in postsynaptic muscle increases larval movement while the reexpression in other tissues had no effect. How does that support the conclusion? Also, why is the control group KisLM27/Kisk13416 missing in this graph? It is very confusing to have to jump to the controls of Figures 8 and 9 first to understand Figure 7D.

We thank the reviewer for pointing out this discrepancy in the text. We have amended Figure 7 to include the UAS-Kis-L data (see 8 below) and the kisLM27/kisk13416 mutant along with its control, w1118. We did not include kisLM27/kisk13416 as a control for statistical comparisons, however, because the transheterozygous mutant kisLM27/kisk13416 is not a control for the reexpression experiments. Reexpression of Kis requires introducing the UAS-kis-L transgene in the kisLM27 mutant background and introducing the Gal4 drivers, which are transgenes, in the kisk13416 background. The chromosomal location of the transgenes can affect phenotypes depending on their respective insertion sites in the genome and there can be “leaky” expression of transgenes 4,5. Therefore, given that the presence of transgenes in the genome may produce phenotypes 4,5, outcrossed controls (the kis stocks including either the UAS or the Gal4 crossed with w1118 controls) represent the closest isogenic controls for kis mutants expressing Kis in a specific tissue.

Our data shows that reexpression of Kis in all tissues but not presynaptic motor neurons or postsynaptic muscle increases the distance traveled indicating that kis expression in all tissues is important for larval locomotion. We have amended the text to indicate this and thank the reviewer for catching this error.

8) Line 369: UAS-kis-L data should be shown.

These data have been added to Figure 7.

9) Line 451: Please show the unpublished data as supplement

These data are included in another manuscript and, therefore, cannot be shown here. We amended the text to omit the information that referenced “data not shown”.

Reviewer #2: This paper extends prior work from the lab showing roles for CHD genes in endocytosis by examining the role of the Drosophila kismet in synaptic vesicle recycling. The authors report differences in both clathrin mediated endocytosis and a backup endocytic pathway, activity dependent bulk endocytosis and then use tissue-specific rescue and knock down experiments to try to tease out the where kismet is required for its functions, with results pointing to a role in post-synaptic muscles. Further ATPase- mutants of kismet rescues both endocytosis and movement dysfunction in the mutant whereas human CHD7 only rescues movement functions. Overall, the study adds to our understanding of this large protein and its novel role in synapses. The discussion is thought provoking and contextualizes the work.

Prior to publication, I have major issues that need to be addressed, most important of which is related to Figures 4 and 5 and the use of drugs and their interpretation:

In figure 4, stars for significance are hard to distinguish from plotted data.

We have moved the histograms for Figures 4 and 5 below the panels showing the raw data to address the comment by the third reviewer. We agreed that the histograms were too small making it difficult to discern the stars for significance and read the text of the graphs.

The quantification in the graphs does not appear to match the images shown, which present a much greater than 2-fold reduction in intensity. More representative average images should be shown. 

All images to construct the figures were selected because the image was the closest to the mean relative FM 1-43FX intensity. We have amended the figure by choosing different representative images for some genotypes and conditions.

Furthermore it is unclear how the ratios are assessed here. They state that they are normalizing all of the samples to w1118 but the exposure cases should be normalize to the vehicle control to determine if the pre-treatment has an effect. In that case, it would appear (from the images provided) that kis is affected by Roscovitine but not EGTA or BAPTA, but dap is only affected by BAPTA and for Iqf and sgg it is unclear because the controls are already very weakly green. The interpretation that this shows kis affects just ADBE seems unjustified. I have similar concerns about the images in Figure 5 and the quantification.

We included additional graphs to show the data normalized to the vehicle control for each genotype. These data were requested by Reviewer 1 and are shown in Figure S3. Our interpretation is that Kis affects both ADBE and CME as evidenced by kis mutants not exhibiting further reductions in endocytosis when BAPTA and EGTA were used to inhibit ADBE and when Chlorpromazine and Dynasore were used to inhibit CME. These data are now shown in Figures 4, 5 and S3.

More detail should be provided about the molecular nature of each inhibitor and what it does. It is unclear why different mutants would respond differently to inhibitors of the same process.

Although each of these inhibitors were previously used to inhibit ADBE and/or CME, there is no indication in the literature why the inhibitors would differentially affect the same genotypes. We could speculate that the affinity of the inhibitor for its molecular target and the penetrance of the mutant alleles may contribute to these differential affects. 

Fig 2. BAPTA and Dynasore are used on the controls, but only the latter on the mutants to show involvement in CME. Both should be tested on the mutants to corroborate the the findings.

We think the reviewer is referring to Fig S2, which shows that none of the compounds used, BAPTA, Dynasore, nor EGTA impair endocytosis in w1118 controls. Endocytosis was induced by high frequency stimulation (20 Hz for one minute) and evoked eEJC amplitudes were recorded. Because none of the compounds impaired endocytosis in controls, they were not used on the mutants.

In figure 2A, the curves for kis and Iqf practically overlap but it is shown that only kis mutants are statistically different than control. This is not at all apparent why this would be the case, especially since we do not see error bars for each time point. I think expanding the Y axis in the region from 50-85 would be helpful to see the differences that are being claimed.

We appreciate this suggestion and made several iterations of this figure prior to submission. Because of the proximity between the curves, adding error bars made it impossible to distinguish one line from another. It was also difficult to expand the Y-axis in the region specified as it distorted the rest of the graph. We revised the Figure so that the 2A panel is larger. This makes it easier to delineate the curves relative to one another.

Again in figure 7, it seems to me that the statistical differences should be tested between the kis heterozygote and homozygote with the gal4 tissue-specific driver not compared with the UAS-kis-L pan driver. It is also unclear why the kis/+; 24B-GAL4/+ has reduced endocytosis in the first place. It suggests that there is a defect in endocytosis caused by the Gal4 drives itself.

We appreciate these suggestions. We did not statistically compare kisLM27/kisk13416 mutants with kisk13416 homozygotes with Gal4 drivers partly because the stocks containing Gal4 drivers are not homozygous for the Gal4 drivers. In addition, both UAS-kis-L and the Gal4 drivers are transgenes. As described above (Reviewer 1 #7), transgene insertion sites can affect phenotypes depending on their respective insertion sites in the genome. Further, there can be “leaky” expression of transgenes 4,5 and these potential influences on phenotypes are not present in kisLM27/kisk13416 mutants. Therefore, given that the presence of transgenes in the genome may produce phenotypes 4,5, outcrossed controls (the kis stocks including either the UAS or the Gal4 crossed with w1118 controls) represent the closest isogenic controls for kisLM27/kisk13416 mutants expressing Kis in a specific tissue.

Minor details that should be addressed:

It should be stated that sgg is on the X chromosome necessitating the use of females since non-fly readers do not know that the first chromosome is the X.

This is a good point. The manuscript includes that information in the first section of the methods.

Significance is not marked in Figure 1. Are any significantly different from controls?

Statistical analyses are problematic with n<5 6. All RT-qPCR experiments included 3-4 biological replicates. Each biological replicate includes 30 central nervous systems. Three technical replicates were performed for each biological replicate. Because all three technical replicates used the same pool of RNA, they aren’t considered independent samples. Therefore, we have not previously published statistical analyses of our RT-qPCR data 2,7 and this is consistent with recent publications that also show RT-qPCR data but do not use t-tests to analyze those data 8,9. 

It is stated that “Recovery from HFS was impaired in kis mutants from 40-50 seconds after HFS 240 and at 40 seconds after HFS in lqf mutants,” but the significance starts show differences at 25s and 30s and respectively.

Recovery from high frequency stimulation (HFS) occurs after the HFS stimuli ceased. Thus, the significant differences at 25 and 30 sec occurred during HFS. Once the HFS stimuli ceased, both kis and lqf mutants exhibited impaired recovery as evidenced by diminished evoked eEJCs at 40-50 and 40 sec, respectively, after HFS. We added a phrase in the text to clarify this.

Line 358, “Similary” should be “Similarly”

Line 426 “for both endocytosis”. Only one item is listed. Remove “both”

Line 437. I think you mean Figs 4,5 not 5,6

Line 480-481: “This circuit may be initiated independent of, but is modulated by, sensory feedback”. needs punctation as marked

We thank the reviewer for finding these typos. We have corrected them.

Are scale bars the same for all images in Figs 4,5, 7? Were these slides imaged with identical settings?

Yes, the scale bars indicate 5 μm. Slides/larvae were imaged by taking the mean of settings used for all controls and applying those means to image all experimental animals on the same day. This is described in the methods section under “Immunocytochemistry and FM Labeling”.

Reviewer #3: In this study, Hendricks and Liebl assess roles for Drosophila Kismet, a chromodomain family (CHD) protein, in regulating presynaptic endocytosis. Alleles of the mammalian homologs of Kis (CHD7 and CHD8) are linked to autism spectrum disorders and CHARGE syndrome, and are thought to regulate a variety of cell functions including cell adhesion and endocytosis, possibly through chromatin remodeling and/or transcriptional regulation. Using reduction-of-function alleles, the authors report that Kis participates in clathrin-mediated and clathrin-independent endocytosis. Surprisingly, its expression in postsynaptic cells appears to be responsible for its role in presynaptic endocytosis, although the mechanisms are currently unclear.

The study itself presents novel and interesting observations for roles of Kis in endocytosis and, even though they do not yet hint at the mechanism of action, this work should be a relevant beginning point for future work examining relationships between CHD proteins, endocytosis and/or regulation of the synaptic vesicle pool, and neuronal function. In places, the authors report counterintuitive findings but do not provide explanations, and as a result many aspects of this manuscript were difficult to follow. The manner in which data were presented in the figures also created challenges.

Specific comments:

1. Many of graphs were very difficult to interpret when printed at page size. Making the graphs larger or increasing font size of axis labels would be helpful.

We appreciated this suggestion and revised Figures 4 and 5. We also revised Figures 2, 3, and 7 to make graphs larger.

2. Throughout the manuscript, the authors distinguish between two forms of endocytosis: clathrin-mediated (CME) and activity-dependent bulk endocytosis (ADBE, which is clathrin-independent). Even in the abstract, the authors state that endocytosis occurs via these two mechanisms, but this is an oversimplification as there are numerous clathrin-independent endocytic pathway. At neurons, ultrafast endocytosis is a clathrin-independent pathway that is distinct from ADBE and which appears to play a prominent role in replenishment of the SV pool. Even if the focus of this study is on CME versus ADBE, the authors should broaden their discussion of endocytic pathways throughout the manuscript to include the likelihood that other pathways are also involved.

Although we acknowledge that other forms of endocytosis occur at synapses, we only alluded to this in the introduction and discussion. We have amended these descriptions in both locations and revised the abstract to mention the other forms of endocytosis. 

3. In Figure 1, the authors describe changes in expression for genes linked to CME, ADBE, or both using a kis hypomorph (k13416) and a loss-of-function (heterozygous LM27/k13416). They report 50% reductions of 150% increases in expression for some genes compared to WT, but the significance (statistical and biological, especially since transcriptionally inactive Kis remains functional for endocytosis and locomotion in their later analyses) is not clear. Statistical analyses should be performed to compare the LM27/k13416 to WT controls and to the k13416 hypomorph.

We appreciate this suggestion. See our response to reviewer 2 above.

4. The curves in Figs. 2A and 3A, with accompanying explanation, were confusing. In the text (lines 239-40), the authors state that “recovery from HFS was impaired in kis mutants from 40-50 seconds after HFS and at 40 seconds after HFS in lqf mutants”. Later (lines 253-5), the authors state that kis mutants exhibit a more severe reduction than lqf mutants. In both cases, the wording in the text implies differences between the two mutants, but curves for kis and lqf look essentially identical to one another. At the least, the wording is imprecise and should be modified, but as it stands the data do not support the authors’ conclusions. The curves (as well as those in Figs. 2C, S1 and S2) lack error bars that would add context. Related to this, the fact that Lqf (epsin) is directly involved in CME as an adaptor, and its dephosphorylation triggers ADBE, suggests that it plays roles in both processes. As a result, it is difficult to interpret data using lqf mutants in the context of separating roles in CME versus ADBE.

We thank the reviewer for pointing out this wording. We have revised the text so that there are no implied differences between kis and lqf mutant responses to HFS and low frequency stimulation. We chose to use lqf mutants for the very reason the reviewer describes. We hypothesized that Kis may be important for both CME and ADBE. To begin to test this hypothesis, we needed to compare kis mutants to another mutant that also would affect both processes. We chose the lqf mutant because of its roles in CME and ADBE as described in the results. Subsequent experiments using chemical inhibitors and additional mutants were designed to assess either CME or ADBE.

5. The authors use Dynasore to inhibit dynamin, with the aim of blocking CME. In Figures S1 and S2, they report that Dynasore inhibited CME but not ADBE. This is surprising because there is a solid body of evidence in the literature demonstrating that dynamin is involved in both endocytic pathways [e.g., Winther et al. (2013) J Cell Sci 126:1021-31 and others]. The reported effect on CME (Fig. S1) is surprisingly small and not consistent across time. Without error bars on these plots it is really difficult to assess whether there is or is not a difference, and the small magnitude of change is not convincing.

Yes, Dynasore inhibited CME and its effect is quite robust. The most pronounced effect was observed from 160-250 sec. During this time, control eEJCs were 75.6-77.7% of the first response. In contrast, Dynasore treatment produced eEJCs that were 57.8-64.1% of the first response. We added this text to the results. We were also surprised that Dynasore along with BAPTA and EGTA did not affect eEJC amplitudes using a stimulation paradigm that induces ADBE. As described earlier, each of these inhibitors were previously used to inhibit ADBE (see text for references). We used the same pretreatment times and conditions as published protocols but did not observe changes in ADBE.

6. The finding that postsynaptic Kis is required for presynaptic endocytosis is interesting, but counterintuitive. The authors speculate extensively about possible explanations, but do not offer any experimental evidence to support any of these explanations. As a result, these results are descriptive. In the final paragraph of the discussion (lines 533-4), the authors state that “our data begins to uncover potential mechanisms by which aberrant chromatin remodeling affect synaptic processes,” but this study is not mechanistic in nature and this statement should be re-phrased.

We rewrote this sentence to, “Our data begins to uncover potential correlates between aberrant chromatin remodeling and synaptic processes.” We do, however, present experimental evidence to explain why postsynaptic Kis may regulate presynaptic endocytosis. Endocytosis occurs adjacent to cell adhesion molecules 10, which link the cells of the synapse to downstream signaling pathways and their actin cytoskeletons. Kis restricts the localization of the cell adhesion molecule, FasII 11, and this may restrict endocytosis. Further, retrograde signaling impacts many presynaptic processes 12,13.

7. The methods do not adequately describe how FM1-43 uptake was quantified. Specifically, did the authors correct for size of the regions they measured?

See response to Reviewer 1 #3.

8. The graphs for Fig. 4 are missing labels to identify the samples with blue data points.

The labels were present but may have been overlooked due to the size of the graphs as this reviewer pointed out. We have increased the size of the graphs in both Figures 4 and 5.

9. Wording of the text accompanying Fig. 7 was extremely confusing, and the experiments appear to lack some controls. This section may be difficult to follow for someone unfamiliar with Drosophila genetics. The data shown in Fig. 7 (images and graphs) needs to include WT and kis controls as a frame of reference for the re-expression experiments; moreover, a Kis-restored UAS should be included to show results for whole-organism re-expression. Statistical analyses should be performed relative to WT or kis mutant controls, in addition to the driver-specific controls. Why do the authors see significance compared to the UAS control but not compared to the driver-specific control in Fig. 7B?

We also anticipated the challenge for individuals in understanding Drosophila genetics. This is why we included “UAS outcrossed control”, “neuron outcrossed control”, and so on in the panels of the figure. The purpose of the experiments shown in Figure 7 was to identify the tissue-specific requirements for Kis’ role in endocytosis. Given the amount of data already shown in the figure, we chose to omit the restoration of Kis expression in all tissues of kis mutants because those genotypes (one experimental animal and one outcrossed control) do not help identify tissue-specific requirements. w1118 controls and kis mutants are not isogenic controls for kis mutants expressing Kis in specific tissues as described above in the response to Reviewer 1 #7. 

10. On lines 357-8, the authors describe Kis knockdown from a previous study in wing discs, but they should also include confirmation and quantification of knockdown in their experimental setup.

We have added the data for knockdown of Kis in all tissues to the manuscript. We don’t have quantification data, however, for neuron-specific, muscle-specific, or glial-specific knockdown. This is because it is not possible to separate neuronal tissue (including glia) from muscle at the Drosophila NMJ because of the structure of the NMJ. The presynaptic motor neurons innervate muscles by forming boutons between muscles and these boutons are buried within muscle tissue.

11. The finding that transcriptionally-inactive Kis restores endocytosis and locomotor behavior (Fig. 9) is surprising and interesting. Immediately after presenting these results, the first paragraph of the discussion states that “our data indicate that Kis promotes both CME and ADBE, possibly by regulating the expression of gene products that mediate these processes” (lines 424-5). It is true that the authors report changes in gene expression in Fig. 1 (although statistical analyses were not presented as noted above), but this statement disagrees with the data from Fig. 9, which suggest that Kis acts independently of its role in regulating gene expression. There is an entire section of the discussion (lines 487-508) devoted to the possible role of Kis in transcriptional regulation of endocytosis, but this may not be relevant given the authors’ findings.

We thank the reviewer for pointing out these sentences in the text. Initially, these sentences may seem contradictory and we have, therefore, revised them. We hypothesize that Kis directly promotes the expression of gene products required for endocytosis while indirectly affecting the expression and/or localization of other gene products required for endocytosis and/or synaptic function. We demonstrated the former by showing that Kis binds both within the promoter and within 200 bp of the transcription start site of dap160 and endoB 7. Kis may indirectly influence the expression and/or localization of other endocytic gene products via several mechanisms, some of which are described in the discussion. First, chromatin remodeling enzymes indirectly influence covalent posttranslational modifications of DNA by assembling in complexes with histone modifying enzymes 14. Second, Kis may influence the localization of some synaptic proteins because it directly affects the expression of scaffolding proteins and cell adhesion molecules 11. Finally, CHD proteins are localized to nucleus, nucleolus, and cytoplasm where they may have context-dependent functions 15.

References

1 Guan, Z., Quinones-Frias, M. C., Akbergenova, Y. & Littleton, J. T. Drosophila Synaptotagmin 7 negatively regulates synaptic vesicle release and replenishment in a dosage-dependent manner. Elife 9 (2020). https://doi.org:10.7554/eLife.55443

2 Hendricks, E. L., Smith, I. R., Prates, B., Barmaleki, F. & Liebl, F. L. W. The CD63 homologs, Tsp42Ee and Tsp42Eg, restrict endocytosis and promote neurotransmission through differential regulation of synaptic vesicle pools. Front Cell Neurosci 16, 957232 (2022). https://doi.org:10.3389/fncel.2022.957232

3 Heo, K. et al. The Rap activator Gef26 regulates synaptic growth and neuronal survival via inhibition of BMP signaling. Mol Brain 10, 62 (2017). https://doi.org:10.1186/s13041-017-0342-7

4 Evangelou, A. et al. Unpredictable Effects of the Genetic Background of Transgenic Lines in Physiological Quantitative Traits. G3 (Bethesda) 9, 3877-3890 (2019). https://doi.org:10.1534/g3.119.400715

5 Haruyama, N., Cho, A. & Kulkarni, A. B. Overview: engineering transgenic constructs and mice. Curr Protoc Cell Biol Chapter 19, Unit 19 10 (2009). https://doi.org:10.1002/0471143030.cb1910s42

6 de Winter, J. C. F. Using the Student's t-test with extremely small sample sizes. Practical Assessment, Research and Validation 18 (2013). https://doi.org:doi.org/10.7275/e4r6-dj05

7 Latcheva, N. K. et al. The CHD Protein, Kismet, is Important for the Recycling of Synaptic Vesicles during Endocytosis. Scientific Reports 9, 19368 (2019). https://doi.org:10.1038/s41598-019-55900-6

8 Borg, R., Herrera, P., Purkiss, A., Cacciottolo, R. & Cauchi, R. J. Reduced levels of ALS gene DCTN1 induce motor defects in Drosophila. Front Neurosci 17, 1164251 (2023). https://doi.org:10.3389/fnins.2023.1164251

9 Dulac, A. et al. A Novel Neuron-Specific Regulator of the V-ATPase in Drosophila. eNeuro 8 (2021). https://doi.org:10.1523/ENEURO.0193-21.2021

10 Grossier, J. P., Xouri, G., Goud, B. & Schauer, K. Cell adhesion defines the topology of endocytosis and signaling. EMBO J 33, 35-45 (2014). https://doi.org:10.1002/embj.201385284

11 Ghosh, R. et al. Kismet positively regulates glutamate receptor localization and synaptic transmission at the Drosophila neuromuscular junction. PLoS One 9, e113494 (2014). https://doi.org:10.1371/journal.pone.0113494

12 Marques, G. & Zhang, B. Retrograde signaling that regulates synaptic development and function at the Drosophila neuromuscular junction. Int Rev Neurobiol 75, 267-285 (2006). https://doi.org:10.1016/S0074-7742(06)75012-7

13 Suvarna, Y., Maity, N. & Shivamurthy, M. C. Emerging Trends in Retrograde Signaling. Mol Neurobiol 53, 2572-2578 (2016). https://doi.org:10.1007/s12035-015-9280-5

14 Jiang, D., Li, T., Guo, C., Tang, T. S. & Liu, H. Small molecule modulators of chromatin remodeling: from neurodevelopment to neurodegeneration. Cell Biosci 13, 10 (2023). https://doi.org:10.1186/s13578-023-00953-4

15 Alendar, A. & Berns, A. Sentinels of chromatin: chromodomain helicase DNA-binding proteins in development and disease. Genes Dev 35, 1403-1430 (2021). https://doi.org:10.1101/gad.348897.121

---

## [Decision Letter · Decision Letter 1]

20 Nov 2023

PONE-D-23-22620R1The CHD family chromatin remodeling enzyme, Kismet, promotes both clathrin-mediated and activity-dependent bulk endocytosisPLOS ONE

Dear Dr. Liebl,

Thank you for submitting your manuscript to PLOS ONE. The manuscript has been evaluated by the same three experts in the field. Although one reviewer was satisfied with the revisions of the text, two other reviewers indicated that proper controls are still missing in several experiments and that the important statistical analyses are not provided in the manuscript. Therefore, after careful consideration, we feel that the manuscript does not meet PLOS ONE’s publication criteria as it currently stands. However, if you feel that you can do all the requested additional experiments and can perform the missing statistical analyses, you may submit a revised version of the manuscript. Please keep in mind that the same reviewers will be reevaluating your manuscript. Thus, you should focus on carefully addressing all previous and new concerns of Reviewer 2 and Reviewer 3.

With best regards,

Alexander G Obukhov, Ph.D.

Academic Editor

PLOS ONE

Reviewers' comments:

Reviewer's Responses to Questions

**Comments to the Author**

1. If the authors have adequately addressed your comments raised in a previous round of review and you feel that this manuscript is now acceptable for publication, you may indicate that here to bypass the “Comments to the Author” section, enter your conflict of interest statement in the “Confidential to Editor” section, and submit your "Accept" recommendation.

Reviewer #1: All comments have been addressed

Reviewer #2: (No Response)

Reviewer #3: (No Response)

2. Is the manuscript technically sound, and do the data support the conclusions?

Reviewer #1: Yes

Reviewer #2: No

Reviewer #3: Partly

3. Has the statistical analysis been performed appropriately and rigorously? 

Reviewer #1: Yes

Reviewer #2: No

Reviewer #3: No

4. Have the authors made all data underlying the findings in their manuscript fully available?

Reviewer #1: Yes

Reviewer #2: Yes

Reviewer #3: Yes

5. Is the manuscript presented in an intelligible fashion and written in standard English?

Reviewer #1: Yes

Reviewer #2: Yes

Reviewer #3: Yes

6. Review Comments to the Author

Reviewer #1: (No Response)

Reviewer #2: In general, I feel that the authors minimize a number of the concerns that the reviewers raised initially:

Multiple reviewers bring up the point that dynasore does not affect ADBE. While the authors state that “they are also surprised”, they do nothing to address this concern except state that they followed published protocols. This is not acceptable. They need to resolve why they are getting different results… this aloofness calls into question their experimental findings as a whole.

Rev #1 comment 6. They also do not address the concern here about comparison to the DMSO control.

The response to Rev #2 Fig S2 is not clear. In this case they say there are no effects to ANY of the compounds but Dyasore is used on the mutants. This needs to be clarified.

Rev 2 comment about molecular nature of compounds. These discussion points need to be added to the text.

BoAll reviewers had issues with controls for Fig 7. They must be compare with the Gal4 driver. While the reviewers consider the outcrossed control the most isogenic, their statements about effects of insertion sites of transgenes, further argues that they data must be compared to the Gal4 driver.

In addressing the need for error bars on the qRT-PCR data, the authors now reveal that the results are with the same pool of RNA. This should actually be done as biological and technical replicates.

Reviewer #3: In their revised manuscript, the authors have made a number of substantial improvements in response to the reviewers’ initial comments and concerns. However, many of the concerns raised were not addressed, and significant issues remain:

1. In my original point 3, I expressed concern about analysis of gene expression in Figure 1, and other reviewers expressed concerns about this figure as well. Reviewer 1 asked about differences in the number of data points, which the authors explain as differences in the number of biological replicates (n=3-4). It is not clear why there are differences in the number of biological replicates across samples. To account for day-to-day differences in extraction efficiency and other experimental variables, and to allow comparisons between relative changes, it would seem to make more sense that the analyses be paired and have the same number of biological replicates where each replicate was prepared side-by-side.

Reviewers 2 and 3 requested statistical analysis for the data in Figure 1, which the authors rebutted in saying that statistical analyses are problematic with n<5. A simple solution to addressing this concern is to perform additional replicate experiments so that their n-value is at least 5, which the authors declined to do. Notwithstanding the issue of having different numbers of biological replicates, which in itself is problematic for the reasons described above, the authors should have repeated this experiment or added the necessary additional trials to the existing data, and performed the requested statistical analyses. As it stands, the authors should not be drawing any conclusions about changes in gene expression if those changes are not supported by rigorous analysis.

2. In my original point 4, I expressed concern about the curves shown in Figs. 2A and 3A. One of the original concerns was that there were no error bars provided (as well as in other curves, e.g., Figs. 3C, 6A, 6B, S1, and S2), and these were not included in the revised manuscript. Reviewer 1 also commented on the lack of error bars for these experiments. Part of the importance of including them is that significance is reported for a very limited number of time points, often with surrounding times that have no reported significance. An alternative interpretation of these results might be that overall recovery is “on the cusp” of being statistically significant, where random noise pushes some time points toward being significant (or pushes some time points toward being not significant). This raises questions about whether there are overall changes in the curves. Regardless, error bars or confidence ranges need to be provided for all curves.

3. In my original point 5, I again expressed concern about the lack of error bars in figures S1 and S2, where the close proximity of data points made it difficult to assess the significance of relative changes. This point was not addressed.

4. In my original point 9, I commented that additional control images and quantification needed to be included, but they were not. The authors stated that they chose to omit the data for restoration of Kis expression in all tissues because of the amount of data already included in the figure; some of these could have been placed in a supplemental figure, especially if the authors did not feel that the data helped identify tissue-specific requirements.

5. In my original point 10, I requested confirmation of knockdown in the experiments performed. The authors responded that they added the data for knockdown in all tissues (reported in the text as a 56.2% reduction), but this should be listed as “data not shown” because they did not actually show the result. These data should be added to Figure 7.

7. PLOS authors have the option to publish the peer review history of their article (what does this mean?). If published, this will include your full peer review and any attached files.

Reviewer #1: No

Reviewer #2: No

Reviewer #3: No

---

## [Author Response · Author response to Decision Letter 1]

14 Dec 2023

We thank the reviewers for reviewing our manuscript for a second time. We have addressed each of the points below by providing additional explanation, justification based on published literature, and highlight revisions to the manuscript. Reviewer comments are shown in normal text. Our responses are in blue.

Reviewer #2: In general, I feel that the authors minimize a number of the concerns that the reviewers raised initially:

Our intention was not to give the impression that we were minimizing the concerns of the reviewers. Instead, we provided a rationale for the results and representation of data. We often refer to published literature to emphasize that we are not arbitrarily making decisions about our results and representation of data. Instead, we consult the literature to ensure we don’t deviate from recent published standards. 

Multiple reviewers bring up the point that dynasore does not affect ADBE. While the authors state that “they are also surprised”, they do nothing to address this concern except state that they followed published protocols. This is not acceptable. They need to resolve why they are getting different results… this aloofness calls into question their experimental findings as a whole.

We did not mean to give the impression that we are lackadaisical but we attributed potential differences to the lack of more literature using Dynasore in flies and/or the challenge of data reproducibility. There are published reports that Dynasore inhibits activity-dependent bulk endocytosis (ADBE) in mammalian in vitro experiments as evidenced by a decrease in dextran import in cultured hippocampal [1] and cerebellar granule neurons [2]. The reports of Dynasore use in flies are largely limited to clathrin-mediated endocytosis [3, 4] but Dynasore was shown to inhibit ADBE in adult fly brains as indicated by inhibited dextran import in surface glia [5]. The challenge of reproducing data has been recognized in the scientific community for over a decade [6]. We strive to mitigate this challenge by ensuring biological replicates represent the variation that is inherent in any phenotype [7].

Rev #1 comment 6. They also do not address the concern here about comparison to the DMSO control.

Reviewer 1’s comment was, “The presentation of the data (graphs) is very confusing. Here, the untreated and treated genotypes need to be compared to assess how strong the effect of the treatment is in WT and mutants and to judge if the treatments have any effect compared to the respective DMSO control.” This comment was made in reference to Figures 4 and 5. Therefore, in response to Reviewer 1’s comment, we added Fig S3, which shows each condition compared to its DMSO control for each genotype. The data corresponding to Figures 4 and 5, then, are shown both on a compound by compound basis (Fig 4, 5) and a genotype by genotype basis (Fig S3).

The response to Rev #2 Fig S2 is not clear. In this case they say there are no effects to ANY of the compounds but Dyasore is used on the mutants. This needs to be clarified.

Fig S2 specifically assesses ADBE, not clathrin mediated endocytosis (CME). Dynasore was only used in experiments to inhibit CME (Fig 3) because we showed in Fig S1 that Dynasore impairs CME induced by 5 Hz stimulation in HL-3 containing 1.0 mM Ca2+. This is specified in the results, “We also used Dynasore, an inhibitor of Dyn GTPase activity [8], to block CME.”

Rev 2 comment about molecular nature of compounds. These discussion points need to be added to the text.

We added additional description of each compound in the text of the results. 

BoAll reviewers had issues with controls for Fig 7. They must be compare with the Gal4 driver. While the reviewers consider the outcrossed control the most isogenic, their statements about effects of insertion sites of transgenes, further argues that they data must be compared to the Gal4 driver.

We apologize if we’re misinterpreting the reviewer’s comment but each of the Gal4 driver controls are included in Figure 7. The Gal4 driver controls are compared with the tissue-specific experimental animals where the Gal4 driver is used to express a UAS transgene. Although some researchers only show Gal4 or UAS outcrossed controls, we show both. The latter is consistent with most other Drosophila publications using the Gal4-UAS system (for recent examples see [9-12]) and mitigates the concern of transgene-specific effects. We did not compare the outcrossed controls with w1118 or the wild type Drosophila strains, OR or wt-B, because those comparisons do not answer the experimental questions posed. Further, we could not find any examples of statistical comparisons between Gal4 or UAS outcrossed controls with w1118, OR, or wt-B, in the published literature where the goal of the experiment was to uncover tissue-specific expression requirements.

In addressing the need for error bars on the qRT-PCR data, the authors now reveal that the results are with the same pool of RNA. This should actually be done as biological and technical replicates.

Our response to this point was, “All RT-qPCR experiments included 3-4 biological replicates. Each biological replicate includes 30 central nervous systems. Three technical replicates were performed for each biological replicate. Because all three technical replicates used the same pool of RNA…” Thus, there were both biological (at least three) and technical replicates (three). Only the technical replicates, not the biological replicates, used the same pool of RNA, which were isolated from the central nervous systems of 30 larvae.

Reviewer #3: In their revised manuscript, the authors have made a number of substantial improvements in response to the reviewers’ initial comments and concerns. However, many of the concerns raised were not addressed, and significant issues remain:

1. In my original point 3, I expressed concern about analysis of gene expression in Figure 1, and other reviewers expressed concerns about this figure as well. Reviewer 1 asked about differences in the number of data points, which the authors explain as differences in the number of biological replicates (n=3-4). It is not clear why there are differences in the number of biological replicates across samples. To account for day-to-day differences in extraction efficiency and other experimental variables, and to allow comparisons between relative changes, it would seem to make more sense that the analyses be paired and have the same number of biological replicates where each replicate was prepared side-by-side.

The 2-ΔΔC(t) method of quantifying and representing RT-qPCR data, which is described in the methods, controls for the daily potentially confounding variables the reviewer mentions. This method first subtracts the cycle threshold (C(t)) value of the target transcript reaction from the C(t) value for GAPDH to obtain ΔC(t) for each transcript. Then, the difference between the control, w1118, and kis mutant ΔC(t)s was calculated. Thus, 2-ΔΔC(t) yields the fold change of target gene expression in kis mutants relative to controls, both of which were normalized to the reference transcript, GAPDH. 2-ΔΔC(t) are only calculated using RNA samples isolated the same day with RT-qPCR reactions executed simultaneously, each using 100 ng of RNA. We have added the former information and a reference to the first paper that described the 2-ΔΔC(t) calculation to the methods. We thank the reviewer for pointing out this needed clarification to the methods.

Reviewers 2 and 3 requested statistical analysis for the data in Figure 1, which the authors rebutted in saying that statistical analyses are problematic with n<5. A simple solution to addressing this concern is to perform additional replicate experiments so that their n-value is at least 5, which the authors declined to do. Notwithstanding the issue of having different numbers of biological replicates, which in itself is problematic for the reasons described above, the authors should have repeated this experiment or added the necessary additional trials to the existing data, and performed the requested statistical analyses. As it stands, the authors should not be drawing any conclusions about changes in gene expression if those changes are not supported by rigorous analysis.

We did not perform additional biological replicates because the mean number of biological replicates for RT-qPCR experiments is three. However, we recognized that this mean may have changed since we last published RT-qPCR data and, therefore, searched recent literature for neuronal gene expression RT-qPCR data. Although there are examples in the literature of using two biological replicates (with three technical replicates per biological replicate), these are likely due to the scarcity of vertebrate tissue samples (see for example [13]). All other recent publications we examined used three biological replicates. We did, however, find one paper that used four biological replicates with three technical replicates per biological replicate [14]. Therefore, we added an additional biological replicate to our data and have amended Figure 1 to show this. We also found examples of statistical analyses of RNA fold changes [13, 15] and have added this to Figure 1. 

2. In my original point 4, I expressed concern about the curves shown in Figs. 2A and 3A. One of the original concerns was that there were no error bars provided (as well as in other curves, e.g., Figs. 3C, 6A, 6B, S1, and S2), and these were not included in the revised manuscript. Reviewer 1 also commented on the lack of error bars for these experiments. Part of the importance of including them is that significance is reported for a very limited number of time points, often with surrounding times that have no reported significance. An alternative interpretation of these results might be that overall recovery is “on the cusp” of being statistically significant, where random noise pushes some time points toward being significant (or pushes some time points toward being not significant). This raises questions about whether there are overall changes in the curves. Regardless, error bars or confidence ranges need to be provided for all curves.

We apologize for not including error bars in our first revision. As we described, inclusion of the error bars makes it challenging to delineate the different points of the conditions/genotypes. We agree with the reviewer that error bars are important and have revised Figures 2, 3, 6, S1, and S2 to include these data. 

3. In my original point 5, I again expressed concern about the lack of error bars in figures S1 and S2, where the close proximity of data points made it difficult to assess the significance of relative changes. This point was not addressed.

We revised Figures S1 and S2 to include error bars.

4. In my original point 9, I commented that additional control images and quantification needed to be included, but they were not. The authors stated that they chose to omit the data for restoration of Kis expression in all tissues because of the amount of data already included in the figure; some of these could have been placed in a supplemental figure, especially if the authors did not feel that the data helped identify tissue-specific requirements.

We agree that these data are ideal for a supplemental figure and thank the reviewer for offering this suggestion. We included the kis mutant phenotype, restoration of Kis in all tissues of kis mutants, and the RT-qPCR data requested in 5 below in a new supplemental figure, S4 Fig.

5. In my original point 10, I requested confirmation of knockdown in the experiments performed. The authors responded that they added the data for knockdown in all tissues (reported in the text as a 56.2% reduction), but this should be listed as “data not shown” because they did not actually show the result. These data should be added to Figure 7.

These data are included in S4 Fig.

1. Li YY, Zhou JX, Fu XW, Bao Y, Xiao Z. Dephospho-dynamin 1 coupled to activity-dependent bulk endocytosis participates in epileptic seizure in primary hippocampal neurons. Epilepsy Res. 2022;182:106915. Epub 20220330. doi: 10.1016/j.eplepsyres.2022.106915. PubMed PMID: 35390701.

2. Clayton EL, Anggono V, Smillie KJ, Chau N, Robinson PJ, Cousin MA. The phospho-dependent dynamin-syndapin interaction triggers activity-dependent bulk endocytosis of synaptic vesicles. J Neurosci. 2009;29(24):7706-17. doi: 10.1523/JNEUROSCI.1976-09.2009. PubMed PMID: 19535582; PubMed Central PMCID: PMCPMC2713864.

3. Gagliardi M, Hernandez A, McGough IJ, Vincent JP. Inhibitors of endocytosis prevent Wnt/Wingless signalling by reducing the level of basal beta-catenin/Armadillo. J Cell Sci. 2014;127(Pt 22):4918-26. Epub 20140918. doi: 10.1242/jcs.155424. PubMed PMID: 25236598; PubMed Central PMCID: PMCPMC4231306.

4. Nemetschke L, Knust E. Drosophila Crumbs prevents ectopic Notch activation in developing wings by inhibiting ligand-independent endocytosis. Development. 2016;143(23):4543-53. doi: 10.1242/dev.141762. PubMed PMID: 27899511.

5. Artiushin G, Zhang SL, Tricoire H, Sehgal A. Endocytosis at the Drosophila blood-brain barrier as a function for sleep. Elife. 2018;7. Epub 20181126. doi: 10.7554/eLife.43326. PubMed PMID: 30475209; PubMed Central PMCID: PMCPMC6255390.

6. Reproducibility and Replicability in Science. Washington (DC)2019.

7. Voelkl B, Altman NS, Forsman A, Forstmeier W, Gurevitch J, Jaric I, et al. Reproducibility of animal research in light of biological variation. Nat Rev Neurosci. 2020;21(7):384-93. Epub 20200602. doi: 10.1038/s41583-020-0313-3. PubMed PMID: 32488205.

8. Kirchhausen T, Macia E, Pelish HE. Use of dynasore, the small molecule inhibitor of dynamin, in the regulation of endocytosis. Methods Enzymol. 2008;438:77-93. doi: 10.1016/S0076-6879(07)38006-3. PubMed PMID: 18413242; PubMed Central PMCID: PMCPMC2796620.

9. Grice SJ, Liu JL. Motor defects in a Drosophila model for spinal muscular atrophy result from SMN depletion during early neurogenesis. PLoS Genet. 2022;18(7):e1010325. Epub 20220725. doi: 10.1371/journal.pgen.1010325. PubMed PMID: 35877682; PubMed Central PMCID: PMCPMC9352204.

10. Walkowicz L, Krzeptowski W, Krzeptowska E, Warzecha K, Salek J, Gorska-Andrzejak J, et al. Glial expression of DmMANF is required for the regulation of activity, sleep and circadian rhythms in the visual system of Drosophila melanogaster. Eur J Neurosci. 2021;54(5):5785-97. Epub 20210317. doi: 10.1111/ejn.15171. PubMed PMID: 33666288.

11. Wang X, Davis RL. Early Mitochondrial Fragmentation and Dysfunction in a Drosophila Model for Alzheimer's Disease. Mol Neurobiol. 2021;58(1):143-55. Epub 20200909. doi: 10.1007/s12035-020-02107-w. PubMed PMID: 32909149; PubMed Central PMCID: PMCPMC7704861.

12. Zhao B, Sun J, Zhang X, Mo H, Niu Y, Li Q, et al. Long-term memory is formed immediately without the need for protein synthesis-dependent consolidation in Drosophila. Nat Commun. 2019;10(1):4550. Epub 20191007. doi: 10.1038/s41467-019-12436-7. PubMed PMID: 31591396; PubMed Central PMCID: PMCPMC6779902.

13. Greguske EA, Maroto AF, Borrajo M, Palou A, Gut M, Esteve-Codina A, et al. Decreased expression of synaptic genes in the vestibular ganglion of rodents following subchronic ototoxic stress. Neurobiol Dis. 2023;182:106134. Epub 20230424. doi: 10.1016/j.nbd.2023.106134. PubMed PMID: 37100209.

14. McSweeney D, Gabriel R, Jin K, Pang ZP, Aronow B, Pak C. CASK loss of function differentially regulates neuronal maturation and synaptic function in human induced cortical excitatory neurons. iScience. 2022;25(10):105187. Epub 20220923. doi: 10.1016/j.isci.2022.105187. PubMed PMID: 36262316; PubMed Central PMCID: PMCPMC9574418.

15. Tan FHP, Azzam G, Najimudin N, Shamsuddin S, Zainuddin A. Behavioural Effects and RNA-seq Analysis of Abeta42-Mediated Toxicity in a Drosophila Alzheimer's Disease Model. Mol Neurobiol. 2023;60(8):4716-30. Epub 20230505. doi: 10.1007/s12035-023-03368-x. PubMed PMID: 37145377.

---

## [Decision Letter · Decision Letter 2]

29 Jan 2024

PONE-D-23-22620R2The CHD family chromatin remodeling enzyme, Kismet, promotes both clathrin-mediated and activity-dependent bulk endocytosisPLOS ONE

Dear Dr. Liebl,

Thank you for submitting your revised manuscript to PLOS ONE. The manuscript has been reevaluated by two previous reviewers. After careful consideration, we feel that it has merit but needs minor revisions before it can be further considered for publication in PLOS ONE. Therefore, we invite you to submit a revised version of the manuscript that addresses the points raised by Reviewer 3.  

We look forward to receiving your revised manuscript.

Kind regards,

Alexander G Obukhov, Ph.D.

Academic Editor

PLOS ONE

Journal Requirements:

Reviewers' comments:

Reviewer's Responses to Questions

**Comments to the Author**

1. If the authors have adequately addressed your comments raised in a previous round of review and you feel that this manuscript is now acceptable for publication, you may indicate that here to bypass the “Comments to the Author” section, enter your conflict of interest statement in the “Confidential to Editor” section, and submit your "Accept" recommendation.

Reviewer #2: All comments have been addressed

Reviewer #3: (No Response)

2. Is the manuscript technically sound, and do the data support the conclusions?

Reviewer #2: Yes

Reviewer #3: Partly

3. Has the statistical analysis been performed appropriately and rigorously? 

Reviewer #2: Yes

Reviewer #3: Yes

4. Have the authors made all data underlying the findings in their manuscript fully available?

Reviewer #2: Yes

Reviewer #3: Yes

5. Is the manuscript presented in an intelligible fashion and written in standard English?

Reviewer #2: Yes

Reviewer #3: Yes

6. Review Comments to the Author

Reviewer #2: (No Response)

Reviewer #3: In this revision, the authors have better addressed the concerns raised by reviewers in the two previous rounds of submission. The data are thus improved and more convincing. I have two minor comments (points 1 and 2 below) that should be addressed based on the latest round of revisions, as well as a concern (point 3 below) about one of the major conclusions drawn (that Kis effects are due to regulation of gene expression). For this conclusion to be convincing, the authors would need to do additional supporting experiments; instead, softening the conclusion and incorporating description/discussion of alternative interpretations of their data, especially with the ATPase-dead mutant of Kis, is necessary.

1. Lines 229-30 are imprecise: as written, the authors state that PI3K92E levels are increased in kis mutants, but this is true only for the LM27/kisk13416 mutant, and not the k13416 hypomorph. The k13416 strain doesn’t seem to be used anywhere else in the paper, so the rationale for its inclusion in this figure is unclear. Importantly, do these two genotypes have the same impact on movement and endocytosis? If so, the fact that only one has a difference in PI3K92E transcript level would imply that the observed change is not important, even if it is statistically significant in LM27/k13416.

2. Line 525: synd should be removed, since the modified data in Figure 1 no longer show changes in its expression.

3. Overall, the fact that ATPase-dead Kis can still correct movement and endocytosis implies that its role in chromatin remodeling (and therefore in regulating transcription) is not required for its regulation of these phenotypes. The authors pointed out in their first rebuttal that Kis localizes to the nucleus, nucleolus, and cytoplasm. Based on these localizations, the conclusion that roles for Kis in endocytosis are due to transcription (e.g., lines 44, 88-90, 522-44) may be true, but cytoplasmic functions cannot be ruled out and indeed may be important given that direct roles of Kis in chromatin remodeling are not required (based on the ATPase-dead mutant results). As written, the conclusion of transcriptional regulation is too strongly stated, especially in the abstract and introduction. It is suitable to speculate on the possibility of gene expression effects in the discussion, but the authors need to make clear that this is one of several possible explanations, and they should expand on alternative interpretations.

7. PLOS authors have the option to publish the peer review history of their article (what does this mean?). If published, this will include your full peer review and any attached files.

Reviewer #2: **Yes: **Judith L. Yanowitz

Reviewer #3: No

---

## [Author Response · Author response to Decision Letter 2]

6 Feb 2024

Response to Reviewer

We thank the reviewer for taking the time to carefully review our manuscript. We believe we have addressed all the concerns raised by the reviewer as described in the points below.

Reviewer #3: In this revision, the authors have better addressed the concerns raised by reviewers in the two previous rounds of submission. The data are thus improved and more convincing. I have two minor comments (points 1 and 2 below) that should be addressed based on the latest round of revisions, as well as a concern (point 3 below) about one of the major conclusions drawn (that Kis effects are due to regulation of gene expression). For this conclusion to be convincing, the authors would need to do additional supporting experiments; instead, softening the conclusion and incorporating description/discussion of alternative interpretations of their data, especially with the ATPase-dead mutant of Kis, is necessary.

1. Lines 229-30 are imprecise: as written, the authors state that PI3K92E levels are increased in kis mutants, but this is true only for the LM27/kisk13416 mutant, and not the k13416 hypomorph. The k13416 strain doesn’t seem to be used anywhere else in the paper, so the rationale for its inclusion in this figure is unclear. Importantly, do these two genotypes have the same impact on movement and endocytosis? If so, the fact that only one has a difference in PI3K92E transcript level would imply that the observed change is not important, even if it is statistically significant in LM27/k13416.

We have amended the text to clarify which specific mutants show significant changes in expression. We also show the kisk13416 data for two reasons. First, kisk13416 is historically the most common kis allele used in publications, possibly because it is homozygous viable. Thus, once published, these data may allow other researchers to draw more direct comparisons with a phenotype of interest. Second, like kisLM27/kisk13416 mutants, kisk13416 mutants also exhibit deficits in endocytosis and reduced synaptic levels of the endocytic proteins Dap160 and EndoB. There are differences, however, in the relative localization of Dynamin (Dyn) to the active zone protein, Bruchpilot (Brp) between kisk13416 and kisLM27/kisk13416 mutants. At rest, Dyn is localized to active zones but is clustered at periactive zones after synaptic stimulation [1]. This relocalization of Dyn does not occur kisk13416 mutants while kisLM27/kisk13416 mutants show a reversal of this relative distribution. Dyn is closer to periactive zones at rest in kisLM27/kisk13416 mutants and then is localized near active zones after stimulation [2]. Collectively, our data indicate that the kisLM27/kisk13416 and kisk13416 mutants have similar phenotypes but there are subtle differences between mutants in other phenotypes.

2. Line 525: synd should be removed, since the modified data in Figure 1 no longer show changes in its expression. 

We removed synd. We thank the reviewer for catching this error.

3. Overall, the fact that ATPase-dead Kis can still correct movement and endocytosis implies that its role in chromatin remodeling (and therefore in regulating transcription) is not required for its regulation of these phenotypes. The authors pointed out in their first rebuttal that Kis localizes to the nucleus, nucleolus, and cytoplasm. Based on these localizations, the conclusion that roles for Kis in endocytosis are due to transcription (e.g., lines 44, 88-90, 522-44) may be true, but cytoplasmic functions cannot be ruled out and indeed may be important given that direct roles of Kis in chromatin remodeling are not required (based on the ATPase-dead mutant results). As written, the conclusion of transcriptional regulation is too strongly stated, especially in the abstract and introduction. It is suitable to speculate on the possibility of gene expression effects in the discussion, but the authors need to make clear that this is one of several possible explanations, and they should expand on alternative interpretations. 

Chromatin remodeling enzymes, like Kis, regulate transcription through more than just the ATPase domain. We have added two paragraphs to the discussion to better describe transcriptional regulation by CHD7, CHD8, and Kis. We better emphasized the importance of chromatin remodeling enzymes as part of multiprotein transcriptional complexes and the functional importance of CHD protein domains and amino acid residues outside of the ATPase domains. Our conclusions in the abstract (line 44) and last paragraph (lines 88-90) of the introduction are focused on the role of Kis in endocytosis. In both conclusions, we propose that Kis “may” and “possibly” promotes endocytosis through transcriptional mechanisms. Thus, we were careful not to strongly conclude a definitive transcriptional role for Kis in endocytosis. We only have direct evidence for Kis regulating endocytosis via transcriptional regulation, however, as Kis binds to the promoter and transcription start sites of endoB and dap160 [3]. We are also careful to recognize that many of the effects of Kis may be indirect e.g. deficient endocytosis may occur due to increased cell adhesion molecules and/or loss of Rab11 in kis mutants (see paragraphs 4-5 of the discussion). 

Although restoration of endocytosis by expressing a Kis lacking the functional ATPase domain in kisLM27/kisk13416 mutants could indicate a cytoplasmic role for Kis, we have no direct evidence for this possibility. Further, there isn’t evidence that can be gleaned from the literature on chromatin remodeling enzymes. Our first rebuttal letter specified that CHD proteins, not Kis, have been detected in nucleus, nucleolus, and the cytoplasm. To our knowledge, there are two published reports of CHD proteins outside the nucleus. CHD9 was detected in the nucleus, nucleolus, and the cytoplasm of osteogenic MBA-15 cells in vitro [4] and CHD1 is found in the cytoplasm of mitotic cells once the nuclear envelope broke down [5]. We have not detected Kis in the cytoplasm and there is little, if any, in the nucleolus of ventral nerve cord cell bodies or postsynaptic muscles [3]. Although some chromatin modifying enzymes also posttranslationally modify cytoplasmic proteins, these enzymes include family members that are primarily localized to the cytoplasm or shuttle between the cytoplasm and nucleus. In contrast, multiple studies demonstrate that point mutations outside the ATPase domain of the Kis homologs, CHD7 and CHD8, produce neuronal transcriptional changes. These data indicate that the transcriptional activity of CHD proteins is not restricted to the ATPase domain. Further, inhibition of KDM5 demethylase activity in Chd8 conditional knock outs partly restores oligodendrocyte precursor cell differentiation [6] indicating that CHD8 partly influences oligodendrocyte precursor cell differentiation by its binding to histone methyltransferases. We have carefully reviewed this information and include additional supporting information in the discussion (see lines 524-569). 

1. Winther AM, Vorontsova O, Rees KA, Nareoja T, Sopova E, Jiao W, et al. An Endocytic Scaffolding Protein together with Synapsin Regulates Synaptic Vesicle Clustering in the Drosophila Neuromuscular Junction. J Neurosci. 2015;35(44):14756-70. doi: 10.1523/JNEUROSCI.1675-15.2015. PubMed PMID: 26538647; PubMed Central PMCID: PMCPMC6605226.

2. Latcheva NK, Delaney TL, Viveiros JM, Smith RA, Bernard KM, Harsin B, et al. The CHD Protein, Kismet, is Important for the Recycling of Synaptic Vesicles during Endocytosis. Scientific Reports. 2019;9(1):19368. doi: 10.1038/s41598-019-55900-6.

3. Ghosh R, Vegesna S, Safi R, Bao H, Zhang B, Marenda DR, et al. Kismet positively regulates glutamate receptor localization and synaptic transmission at the Drosophila neuromuscular junction. PLoS One. 2014;9(11):e113494. Epub 20141120. doi: 10.1371/journal.pone.0113494. PubMed PMID: 25412171; PubMed Central PMCID: PMCPMC4239079.

4. Salomon-Kent R, Marom R, John S, Dundr M, Schiltz LR, Gutierrez J, et al. New Face for Chromatin-Related Mesenchymal Modulator: n-CHD9 Localizes to Nucleoli and Interacts With Ribosomal Genes. J Cell Physiol. 2015;230(9):2270-80. doi: 10.1002/jcp.24960. PubMed PMID: 25689118; PubMed Central PMCID: PMCPMC6363339.

5. Stokes DG, Perry RP. DNA-binding and chromatin localization properties of CHD1. Mol Cell Biol. 1995;15(5):2745-53. doi: 10.1128/MCB.15.5.2745. PubMed PMID: 7739555; PubMed Central PMCID: PMCPMC230505.

6. An Y, Zhang L, Liu W, Jiang Y, Chen X, Lan X, et al. De novo variants in the Helicase-C domain of CHD8 are associated with severe phenotypes including autism, language disability and overgrowth. Hum Genet. 2020;139(4):499-512. Epub 20200124. doi: 10.1007/s00439-020-02115-9. PubMed PMID: 31980904.

---

## [Decision Letter · Decision Letter 3]

26 Feb 2024

The CHD family chromatin remodeling enzyme, Kismet, promotes both clathrin-mediated and activity-dependent bulk endocytosis

PONE-D-23-22620R3

Dear Dr. Liebl,

We’re pleased to inform you that your manuscript has been judged scientifically suitable for publication and will be formally accepted for publication once it meets all outstanding technical requirements.

Kind regards,

Alexander G Obukhov, Ph.D.

Academic Editor

PLOS ONE

Reviewers' comments:

Reviewer's Responses to Questions

**Comments to the Author**

1. If the authors have adequately addressed your comments raised in a previous round of review and you feel that this manuscript is now acceptable for publication, you may indicate that here to bypass the “Comments to the Author” section, enter your conflict of interest statement in the “Confidential to Editor” section, and submit your "Accept" recommendation.

Reviewer #3: All comments have been addressed

2. Is the manuscript technically sound, and do the data support the conclusions?

Reviewer #3: Yes

3. Has the statistical analysis been performed appropriately and rigorously? 

Reviewer #3: Yes

4. Have the authors made all data underlying the findings in their manuscript fully available?

Reviewer #3: Yes

5. Is the manuscript presented in an intelligible fashion and written in standard English?

Reviewer #3: Yes

6. Review Comments to the Author

Reviewer #3: (No Response)

7. PLOS authors have the option to publish the peer review history of their article (what does this mean?). If published, this will include your full peer review and any attached files.

Reviewer #3: No

---

## [Editor Report · Acceptance letter]

11 Mar 2024

PONE-D-23-22620R3 

PLOS ONE

Dear Dr. Liebl, 

I'm pleased to inform you that your manuscript has been deemed suitable for publication in PLOS ONE. Congratulations! Your manuscript is now being handed over to our production team.

Kind regards, 

on behalf of

Dr. Alexander G Obukhov 

Academic Editor

PLOS ONE